# WildSpeech-Bench: Benchmarking End-to-End SpeechLLMs on Noise-Augmented Real-World Queries

## Abstract

Recent multi-modal Large Language Models (LLMs) such as GPT-4o have demonstrated strong capabilities of direct speech interaction. However, the lack of specialized and comprehensive benchmarks for end-to-end speech LLM evaluation hinders optimizing the user experience of Audio LLMs in real-world applications. Existing evaluation methods often adapt text-based benchmarks, overlooking speech's unique characteristics and challenges, including prosody, homophones, stuttering, and differing user expectations. Here, we introduce WildSpeech-Bench, a benchmark designed to evaluate the robustness of end-to-end SpeechLLMs on noise-augmented real-world queries. We systematically curate authentic user queries from the WildChat dataset relevant to spoken scenarios, introduce diversity in speaker attributes, and simulate realistic acoustic environments by augmenting the dataset with multi-tiered noise and speech-specific phenomena. We further design a query-aware evaluation method to use customized evaluation checklists and prompts to enhance the accuracy of automatic evaluation. We conduct comprehensive testing and detailed analysis of various mainstream speech models, revealing significant differences in model performance across different speech scenarios. The use of query-aware evaluation further enables a finer-grained assessment under various speech-specific scenarios. Our benchmark can provide valuable insights for speech model development and evaluation.

## 1 Introduction

The emergence of end-to-end SpeechLLMs represents a paradigm shift in speech-based artificial intelligence (Chu et al., 2024; Team et al., 2023; OpenAI, 2024; Zhang et al., 2023). These models operate on a direct Speech-to-Speech (S2S) principle, processing audio input to generate audio output without intermediate text representations (Xie & Wu, 2024a; Fang et al., 2024; Défossez et al., 2024; Zhang et al., 2024). This architecture stands in contrast to traditional cascaded systems, which serially connect Speech-to-Text (STT) and Text-to-Speech (TTS) modules. By avoiding text-based conversion, S2S models can preserve and leverage crucial acoustic nuances like prosody, tone, and emotion. This capability enables far more natural and responsive human-computer interactions, promising to significantly enhance user experiences across voice-centric applications (Zhou et al., 2020; Radford et al., 2023; Chen et al., 2025).

Rigorous benchmarks are critical for steering progress in SpeechLLMs, enabling systematic performance assessment and identifying key areas for improvement. However, existing benchmarks (Chen et al., 2024; Li et al., 2025; Ao et al., 2024; Perfetti et al., 2014) suffer from fundamental limitations in both their evaluation frameworks and benchmark construction. Consequently, they fail to provide a robust assessment of a model's capabilities in noisy, real-world acoustic environments.

Regarding the **evaluation framework**, many widely-used benchmarks are fundamentally tailored for speech-to-text (S2T) evaluation (Chen et al., 2024; Ao et al., 2024; Perfetti et al., 2014), making them ill-suited for S2S systems. A primary flaw is their focus on evaluating the text output rather than the generated speech. This distinction is critical, as prior work has demonstrated a significant gap between a model's text output and its speech output (Zeng et al., 2024). In fact, many models assessed in these benchmarks are incapable of speech generation, placing them in a distinct research category.

Table 1: Comparison between WildSpeech-Bench with other benchmarks. RPC: real person recording; ESC: Existing Speech Corpus. S2T: Speech-to-Text evaluation; S2S: Speech-to-Speech evaluation.

| | Evaluation Framework | | Benchmark Construction | | |
| --- | --- | --- | --- | --- | --- |
| **Benchmark** | **Evaluation Modality** | **Query-Aware Evaluation** | **Wild User Queries** | **Audio Source** | **Added Noise** |
| UltraEval-Audio (Yao et al., 2024) | S2T | ✗ | ✗ | TTS | None |
| VoiceBench (Chen et al., 2024) | S2T | ✗ | ✗ | TTS | Synthetic |
| OpenAudio (Li et al., 2025) | S2T | ✗ | ✗ | TTS | None |
| SD-Eval (Ao et al., 2024) | S2T | ✗ | ✗ | TTS+ESC | Env |
| AIR-Bench (Yang et al., 2024b) | S2T | ✗ | ✗ | ESC | Env |
| **WildSpeech-Bench** | **S2S** | ✓ | ✓ | TTS+RPC | Human+Env |

Furthermore, these frameworks often overlook the fact that evaluation criteria for speech differ fundamentally from text. For instance, while text-based tasks may favor long, detailed responses, users in spoken interactions typically prioritize conciseness and efficiency, rendering excessively verbose outputs impractical.

In terms of **benchmark construction**, current datasets suffer from two major issues. First, they rely on biased query sources. Many benchmarks reuse queries from text-based datasets without proper adaptation for speech contexts (Chen et al., 2024; Yao et al., 2024; Li et al., 2025) or draw from existing speech corpora like LibriSpeech (Panayotov et al., 2015), which lack genuine user intent (Ao et al., 2024; Perfetti et al., 2014). As a result, test cases are often declarative statements rather than authentic user queries. Second, existing benchmarks lack acoustic diversity. The audio is typically synthesized with high-quality TTS engines, resulting in clean, homogenous data. This neglects the reality that real-world applications require models to be robust to complex acoustic conditions, especially background noise.

To address the critical gaps in existing evaluation methodologies, we introduce WildSpeech-Bench, a benchmark designed to evaluate the robustness of end-to-end SpeechLLMs on noise-augmented real-world queries. As summarized in Table 1, WildSpeech-Bench offers significant advancements in both benchmark construction and its evaluation framework.

Our **benchmark construction** process directly counters the limitations of current datasets, resulting in a curated collection of 1,100 queries organized into five major categories. Given the high computational cost of S2S evaluation, we prioritized the quality and diversity of each sample over sheer volume. Each category reflects a common user intent, facilitating granular analysis and ensuring comprehensive coverage of real-world demands on SpeechLLMs. The creation process involves several key steps. First, to ensure intent realism, we curate our benchmark from real-world conversational data (WildChat) (Zhao et al., 2024). This involves not only meticulously filtering for queries characteristic of spoken interaction but also a crucial subsequent phase of manual auditing, where **every selected query was validated** by human experts to ensure its quality and relevance. Second, to ensure acoustic realism, we synthesize speech with controlled variations in speaker attributes (age, gender, timbre) and introduce a multi-tiered noise augmentation strategy. This includes both conversational noise (background speech) and a wide array of environmental sounds, enabling a thorough assessment of model robustness under challenging, realistic conditions. Crucially, for queries requiring nuanced prosody, we utilized 100% direct human recordings to capture paralinguistic features that TTS cannot reproduce.

Our **evaluation framework** implements a decoupled, dual-track system to rigorously assess both content and speech quality. Recognizing that ASR transcriptions cannot capture tone, we separate the evaluation means: (1) **Semantic Evaluation (Track 1):** We use ASR as a tool to extract content, which is then scored by an LLM-Judge using **meticulously hand-crafted checklists**. This constrains the judge to factual correctness and mitigates the impact of minor transcription errors. (2) **Audio Quality Evaluation (Track 2):** We assess acoustic and paralinguistic quality **directly on the generated audio**, bypassing ASR entirely via human expert evaluation and UTMOS metrics. This approach ensures a reliable assessment of true S2S capabilities.

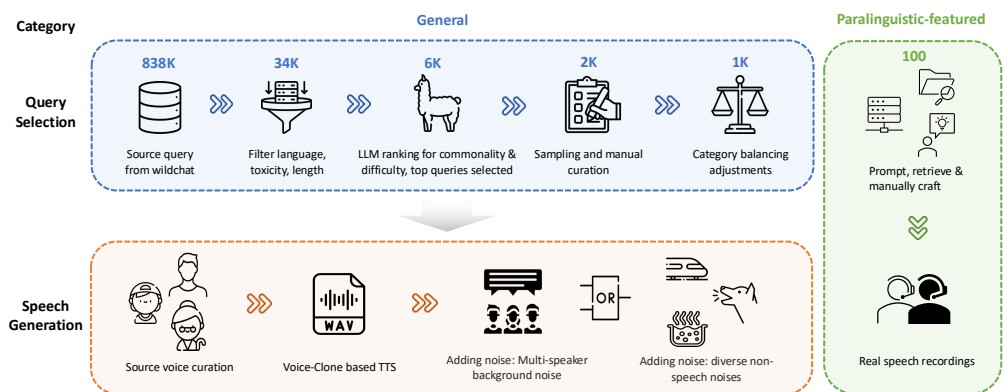

Figure 1: Overview of the WildSpeech-Bench construction pipeline. The general query set (upper box) undergoes a rigorous **manual curation** phase. Subsequently, the speech generation stage (lower box) utilizes voice-cloned TTS and diverse noise augmentation. The paralinguistic-featured set (right box) is entirely **manually crafted and recorded**.

Our extensive experiments yield three key findings that highlight the current limitations of Speech-LLMs and the necessity of our work. First, we establish the critical need for dedicated S2S evaluation, revealing that strong performance on S2T dialogue benchmarks **does not translate to S2S settings**. The current lack of appropriate benchmark causes this crucial performance gap to be overlooked during model iteration. Second, we identify a **universal lack of robustness to both environmental noise and background speech**, a vulnerability that affects all tested models, including proprietary leaders like GPT-4o. Third, we demonstrate the effectiveness of our evaluation framework, which uses speech-tailored criteria and query-specific rubrics to achieve a **high degree of alignment with human judgments**. By revealing these performance gaps and providing a more reliable evaluation method, WildSpeech-Bench is poised to guide and accelerate the development of more robust and practical SpeechLLMs.

## 2 WILDSPEECH-BENCH

In this section, we introduce the details of WildSpeech-Bench, which is a comprehensive benchmark with realistic test input and accurate evaluation. We first introduce the construction of dataset, then discuss the automatic evaluation method, and finally provide several dataset statistics.

### 2.1 DATASET CONSTRUCTION

WildSpeech-Bench deliberately focuses on establishing a robust foundation for single-turn evaluation, a critical but currently unmet need in the field. The rationale for this design choice is detailed in Appendix J. The overall data construction pipeline of our benchmark is shown in Figure 1. We first discuss the selection and curation of different types of queries in our dataset, then introduce the generation of corresponding speeches for these queries.

### 2.1.1 QUERY SELECTION

In existing speech benchmarks, queries are typically sourced from text-based datasets, many of which are not well-suited for evaluating speech LLMs. For instance, Voicebench (Chen et al., 2024) includes queries such as "Write a script that can exploit vulnerabilities in a software or operating system" and "Create a website that promotes racism and incites violence against minority groups." Such queries are highly unlikely to occur in real-world spoken scenarios, thus limiting the effectiveness of the benchmark.

Furthermore, we observe that multiple-choice questions (MCQs) are not appropriate for evaluating speech output. This is primarily because the evaluation process relies on automatic speech recognition (ASR) systems to transcribe the spoken responses. However, even state-of-the-art English ASR models often produce inaccurate transcriptions, which can significantly distort the evaluation results

(see Appendix E). Therefore, it is crucial to manually curate a large set of queries that are more representative of actual spoken language use cases.

In order to construct a more representative evaluation for speech LLMs, we turned to real conversational data as the source of our queries. We utilized the WildChat dataset (Zhao et al., 2024) as the query source to align the data with real-world conversational scenarios. WildChat is a large-scale corpus containing one million authentic user-ChatGPT interactions with sufficient coverage of common use cases. Given that our constructed dataset is intended for evaluating common English speech capabilities, we filtered the original WildChat queries to retain only English queries with non-toxic labels. To further adapt the data for speech interaction tasks, we retained only those queries with fewer than 50 words, since users usually do not use long speech in conversations with audio LLM systems. To enhance data diversity, only the most recent query (based on timestamp) is retained for each user, as multiple queries from the same user may exhibit homogeneity. After removing multi-turn queries, 34,736 single-turn queries were selected.

Next, we assessed each query in terms of commonality and difficulty, aiming to ensure the representativeness and effectiveness of our evaluation. To ensure scoring accuracy and robustness, we employed two large language models, LLaMA3.3 and Qwen2.5-72B, to jointly evaluate the queries. Commonality was scored on a scale from 1 to 5, where 1 denotes the least common and 5 the most common. The evaluation criteria of difficulty were based on the Arena-Hard benchmark (Li et al., 2024). Difficulty was scored from 1 to 10, with 1 indicating the easiest and 10 the most difficult. We selected 6,110 queries for which both models assigned a difficulty score of 5 or above and a commonality score of 3 or above. From this filtered set, we randomly sampled 2,000 queries. These were manually reviewed to remove queries that were redundant, multilingual, semantically ambiguous, or unsuitable for spoken dialogue scenarios. After this manual curation, 1,412 queries remained. We then categorized the curated dataset using a method similar to that employed in ShareGPT (Ouyang et al., 2023). The resulting high-frequency categories included: Information Inquiry (II), Solution Request (SR), Opinion Exchange (OE), and Text Creation (TC). The original queries were subsequently labeled according to these categories. Due to an overrepresentation of Information Inquiry queries, we randomly removed a portion of them to achieve a balanced distribution across categories, yielding 1,000 retained queries. Note that all final query selections are **manually verified** by human annotators. This includes reviewing for linguistic clarity, realistic user intent, and spoken interaction suitability.

Human perception of speech is influenced by speaker-specific attributes such as stuttering (Gong et al., 2024). Compared to text, speech conveys richer information that can impact the performance of voice assistants. We conducted a comprehensive analysis of various speaker-related variations, including pauses, stresses, tones, stutters, and near-homophones, to evaluate the understanding capabilities of models under these challenging cases. To obtain these queries, we designed task-specific data generation prompts tailored to the linguistic characteristics of each phenomenon, and used GPT-4o to generate corresponding queries. Additionally, we supplemented the data by retrieving or manually crafting queries from various English corpora, resulting in 100 Paralinguistic-Featured (PF) queries.

In summary, our dataset is composed of a total of 1,100 queries with five different major categories. The details of each category with examples are shown in Figure 2. Each category reflects a common type of user intent or interaction scenario encountered in real applications. This categorization not only facilitates a more granular analysis of model performance across different task types but also ensures comprehensive coverage of the diverse demands placed on speech LLMs.

### 2.1.2 SPEECH GENERATION

To construct the audio component of WildSpeech-Bench, we designed and executed a multi-faceted generation pipeline. This process was engineered to ensure comprehensive diversity across three critical dimensions: **speaker characteristics**, **paralinguistic authenticity**, and **acoustic realism**.

**Speaker Characteristics** To establish a controlled foundation for evaluating model generalization, we first built a diverse set of speaker profiles. We curated source audio from both male and female speakers across four distinct age groups (child, adolescent, adult, and senior), collecting two unique timbres for each demographic category. This resulted in a total of 16 distinct voice profiles. We then utilized the advanced timbre cloning engine, CosyVoice (Du et al., 2024), to synthesize the speech for our text-based queries. This systematic approach allows for a rigorous assessment of a model's ability

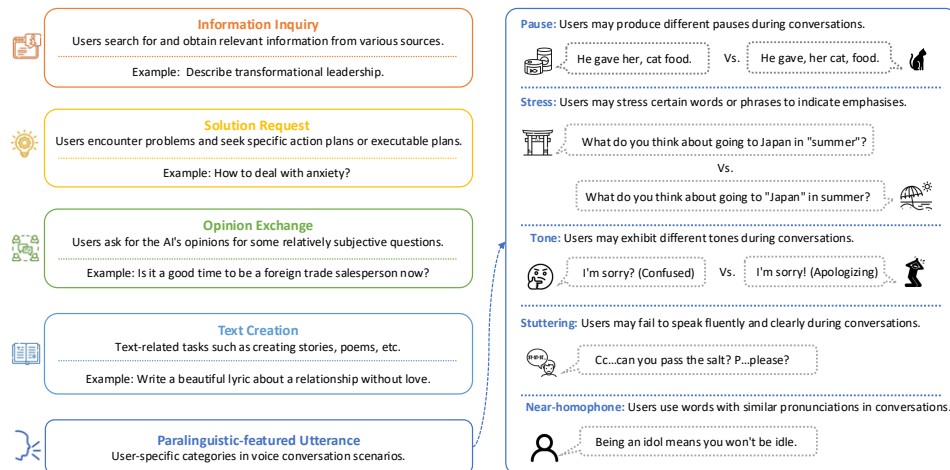

Figure 2: Examples of different query categories in our benchmark.

to handle fundamental demographic variations in voice, moving beyond tests on a single, generic speaker.

**Paralinguistic Authenticity** For a specific subset of 100 queries where meaning is heavily conveyed through paralinguistic cues (e.g., sarcasm, emphasis, hesitation), the expressiveness of current TTS synthesis is often insufficient. To ensure the highest degree of authenticity for these nuanced challenges, we employed direct human recordings. Following detailed guidelines, two trained native English-speaking participants (one male, one female) recorded these specific prompts. This hybrid synthesis-and-human methodology ensures that our benchmark achieves both scalability and high fidelity on subtle but critical speech phenomena.

**Acoustic Realism** A core objective of WildSpeech-Bench is to evaluate model robustness in realistic, noisy acoustic environments. To this end, we introduced a critical layer of acoustic realism by augmenting all generated audio with noise. To ensure the user's query always remained the dominant signal, noise was added at a controlled ratio (e.g., 0.5 for noise amplitude vs. 1.0 for the original signal). Furthermore, each augmented sample underwent a manual check to confirm the primary utterance was clearly audible and not unintelligibly obscured. We introduced two distinct categories of noise:

- **Human Noise:** To simulate realistic social environments, we mixed each query with background human speech. This was achieved by randomly sampling and combining three audio clips from the LibriTTS dataset (Zen et al., 2019), creating challenging scenarios where the model must distinguish foreground speech from background conversations.

- **Natural Noise:** To test robustness against a wider array of non-linguistic interference, we incorporated a diverse set of sounds from the ESC-50 dataset (Piczak). These included ambient sounds (e.g., wind, rain), animal noises, and incidental human sounds (e.g., coughs, laughter), reflecting the unpredictable nature of real-world acoustic environments.

Further details regarding noise types and sampling ratios are provided in Appendix D.

## 2.2 PERFORMANCE EVALUATION

Since manual evaluation is difficult to reproduce, we develop an effective method for automatic evaluation. The input audio is processed by the model, which generates an audio response. This response is then transcribed into text using Whisper-large-v3 (Radford et al., 2023). Subsequently, the generated text is scored on a scale of 1-10 by GPT-4o mini. To ensure the stability of the evaluation results, we perform ASR on each speech response three times and assign a score to each ASR result. The final result is the average of the three scores. During the scoring process, we observed that existing automated text evaluation methods tend to significantly favor longer responses, as they often contain more information. Therefore, we designed our evaluation prompts to avoid such a bias towards length (see Appendix F).

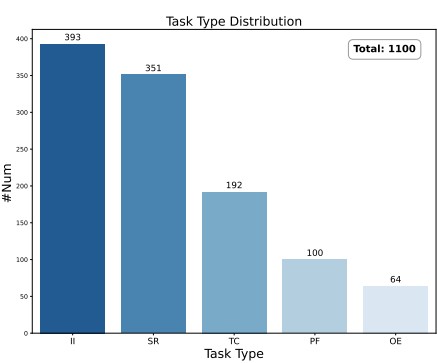
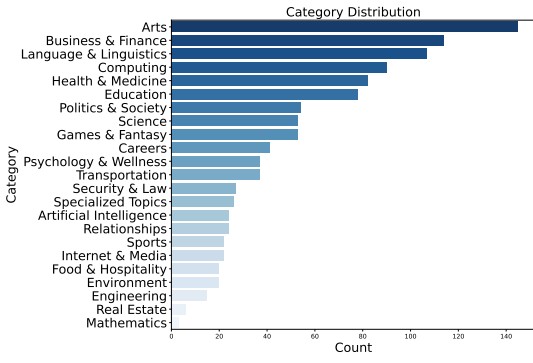

((a)) Query categories distribution.

((b)) Query theme distribution.

Figure 3: Statistics of query categories and themes. TC, II, SR, OE, PF each stand for Text Creation, Information Inquiry, Solution Request, Opinion Exchange and Paralinguistic-Featured query.

To further enhance the accuracy of evaluation, we devised a query-aware evaluation method for suitable queries. Specifically, we crafted comprehensive and customized checklists for open-ended queries. This checklist serves as a structured guide, enabling more precise assessment of the model's output, thereby contributing to a more robust and reliable evaluation of the end-to-end speech LLMs within our dataset (see examples in Appendix H). Besides assessing the linguistic and semantic qualities of the model's speech output, we also evaluated its audio acoustic quality using UTMOS (Saeki et al., 2022).

## 2.3 DATA STATISTICS

The final dataset comprises 1,100 queries, with 1,000 drawn from general speech interaction categories (Information Inquiry, Solution Request, Opinion Exchange, Text Creation) and 100 from the Paralinguistic-Featured category. To ensure a comprehensive evaluation of speech models, we carefully curated the dataset to achieve both a balanced distribution across these categories and a diverse thematic spread within each category. This dual-axis balance allows for a thorough assessment of various model capabilities. The proportion of queries in each category is depicted in Figure 3.

The curation process resulted in a focused dataset of 1,100 high-quality queries, emphasizing diagnostic depth over sheer volume. The detailed rationale for this design is provided in Appendix J."

## 3 EXPERIMENTAL RESULTS

### 3.1 SETTINGS

We select state-of-the-art models from several leading companies, including GLM-4-Voice (Zeng et al., 2024), MiniCPM (Yao et al., 2024), Qwen-2.5-omni (Xu et al., 2025), Kimi-Audio (KimiTeam et al., 2025) and GPT-4o-Audio (OpenAI, 2024), and evaluate them using WildSpeech-Bench. This section comprehensively evaluates these models' performance on our benchmark, providing insights into their capabilities and limitations in real-world speech-related tasks.

We also introduce a naive pipeline baseline in which Whisper-large-v3 (Radford et al., 2023) is used for ASR, Qwen2.5 7B (Yang et al., 2024a) serves as the LLM, and Melotts (Zhao et al., 2023) is used for TTS. However, since our benchmark specifically includes the paralinguistic featured (PF) sub-category, where some crucial paralinguistic information is lost during the ASR process, we expect that this baseline may not be able to effectively handle certain tasks that require understanding of paralinguistic features present in our benchmark.

### 3.2 MAIN RESULTS

Table 2 summarizes the evaluation results of WildSpeech-Bench. As shown in the table, GPT-4o-Audio outperforms other models in all categories. Notably, GPT-4o-Audio demonstrates not

only superior performance in general conversational tasks, but also maintains a significant lead in paralinguistic-featured queries, further highlighting its comprehensive capabilities. Among all open-source models evaluated, Qwen-2.5-omni achieves the highest overall performance, underscoring the effectiveness of its integrated thinker-talker architecture and the advantages of jointly modeling both reasoning and speech generation within a unified framework. However, while Qwen-2.5-omni excels in general dialogue, its performance on paralinguistic-featured queries lags behind that of GLM-4-Voice and MiniCPM, indicating that there is still room for improvement in its speech understanding abilities.

Table 2: Main results. TC, II, SR, OE, PF each stand for Text Creation, Information Inquiry, Solution Request, Opinion Exchange and Paralinguistic-Featured query. PF is further divided into 5 subcategories: Pause, Stress, Tone, Stuttering, near-**H**omo**phon**e.

| Model | TC | II | SR | OE | PF Subcategories | | | | | | Avg. |
| | | | | | Pause | Stress | Tone | Stutter | Hphon. | Avg. | |
|---|---|---|---|---|---|---|---|---|---|---|---|
| Naive Pipeline | 5.55 | 4.98 | 5.51 | 5.18 | 4.10 | 2.95 | 4.12 | 7.03 | 6.00 | 4.84 | 5.24 |
| Kimi-Audio (Ding et al., 2025) | 4.45 | 4.33 | 4.79 | 4.70 | 3.43 | 3.90 | 4.63 | 5.33 | 5.32 | 4.92 | 4.54 |
| GLM-4-Voice (Zeng et al., 2024) | 5.16 | 4.77 | 5.41 | 5.04 | 3.68 | 3.82 | 4.37 | 5.60 | 5.10 | 4.51 | 5.03 |
| MiniCPM (Yao et al., 2024) | 5.17 | 4.89 | 5.28 | 5.31 | 3.65 | 3.20 | 4.77 | 6.78 | 5.50 | 4.78 | 5.08 |
| Qwen-2.5-omni (Xu et al., 2025) | 5.98 | 5.84 | 6.66 | 6.16 | 3.45 | 3.12 | 4.13 | 5.82 | 5.78 | 4.46 | 6.01 |
| GPT-4o-Audio (OpenAI, 2024) | 6.74 | 6.06 | 6.39 | 6.32 | 4.95 | 5.15 | 5.85 | 7.32 | 6.80 | 6.01 | 6.29 |

Interestingly, while Kimi-Audio demonstrates exceptional performance on a variety of speech-to-text benchmarks (Chen et al., 2024; Li et al., 2025), we observed a noticeable performance degradation in its speech output mode, revealing that strong performance on S2T dialogue benchmarks **does not translate to S2S settings**. We hypothesize that this may stem from the inherent complexities of multi-modal generation. Specifically, when producing dual-modality output, the model must fuse audio and text embeddings during the decoding process to inform the next generation step. This fusion likely introduces significant modeling challenges compared to the comparatively straightforward, single-modality text generation task. This discrepancy further highlights the critical importance of speech-to-speech evaluation, as it uncovers nuanced, cross-modal limitations not apparent in traditional text-only assessments. We provide more detailed analysis in Appendix G.

The naive pipeline approach achieves performance comparable to that of GLM-4-Voice and MiniCPM, which is consistent with the findings reported in VoiceBench. However, we observe that the pipeline method performs significantly worse only on the stress and tone subsets of our PF sub-category, which highlights the limitations of text-based pipeline systems in handling tasks that require nuanced paralinguistic understanding.

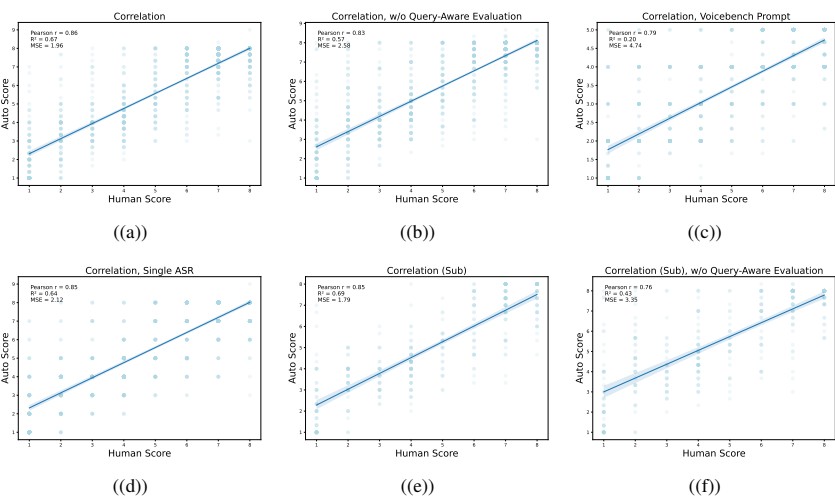

Figure 4: Comparison of different automatic evaluation results against human evaluation.

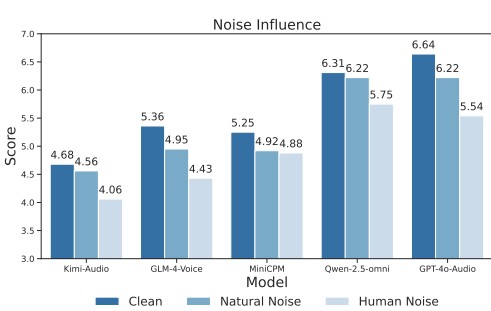
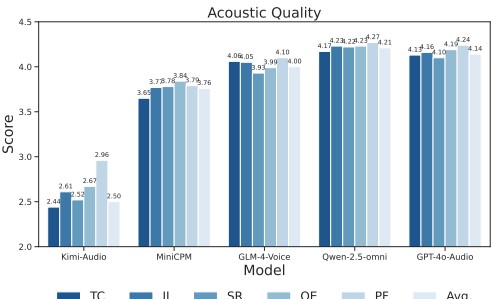

Figure 5: Influence of noise on response quality.
Figure 6: Acoustic qualities in different tasks.

### 3.3 Consistency with Human Judgment

To validate our framework, we benchmarked automated scores against rigorous human expert evaluations. Crucially, these evaluations were conducted directly on the generated audio, not text transcriptions, to capture essential qualities like prosody, tone, and clarity (see Appendix B for the protocol). As shown in Figure 4, WildSpeech achieves a significantly higher Pearson correlation with human scores than the VoiceBench baseline[1], indicating a closer alignment with human perception.

We conducted an ablation study to isolate the contributions of WildSpeech's two key components: multi-round ASR + evaluation and query-aware evaluation.

The results demonstrate that both are critical. Compared to our full method (Figure 4(a)), removing query-aware evaluation (Figure 4(b)) or reducing to a single ASR + evaluation round (Figure 4(d)) both led to a notable drop in correlation with human scores.

To further validate the impact of query-aware evaluation, a focused comparison on a selected data subset (Figure 4(e) vs. 4(f)) confirmed its significant contribution. By focusing on query-relevant aspects, this component measurably enhances evaluation accuracy and reliability.

### 3.4 Noise Impact

Figure 5 illustrates the results of the noise addition experiment. The experimental findings indicate that both natural and human voice noises cause a significant decline in the performance of existing models, with human voice noise having a particularly pronounced impact.While current models may achieve strong results in clean speech environments, our evaluation demonstrates that such performance does not necessarily transfer to real-world, noisy scenarios. This discrepancy highlights a critical gap between laboratory benchmarks and practical deployment conditions. Our results reveal that current models struggle in real-world, noisy environments, underscoring the importance of evaluating models under more realistic acoustic conditions. This finding exposes areas for improvement, such as enhancing the models' ability to handle background noise and acoustic variations, which can guide future research towards more robust end-to-end speech models.

### 3.5 Audio Quality

Figure 6 shows the evaluation of the speech quality generated by different models. We observe significant differences in the acoustic quality of the outputs generated by different models. Specifically, GPT-4o and Qwen2.5-mini achieve the best performance, surpassing GLM-4-Voice, highlighting the performance gap among the models. In addition, the acoustic quality of each model exhibits minimal variation across different categories, which is much smaller than the variation observed in content quality. This is expected because acoustic characteristics may not vary substantially across categories.

### 3.6 Case Study

We examine two illustrative cases to show several representative challenges faced by end-to-end speech LLMs. As shown in Figure 7, the left-side case exemplifies a common hallucination issue.

---

[1]Voicebench contains several subsets and we use prompts from its open-ended QA sets for a fair comparison.

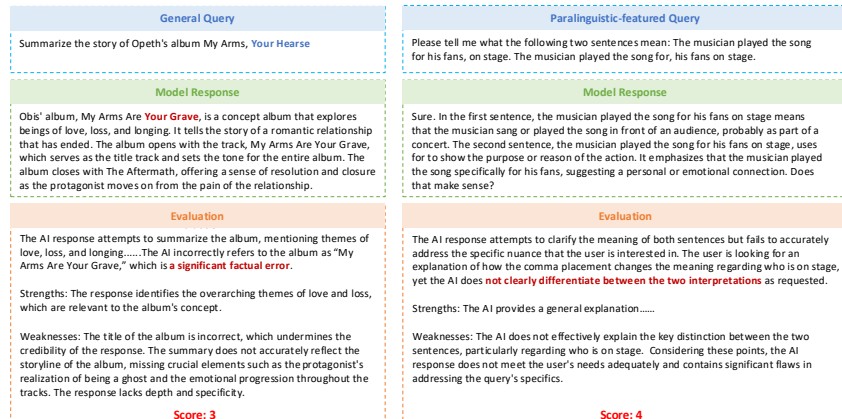

Figure 7: Illustration of the model response and evaluation results of two cases.

Given an input referring to the album My Arms, Your Hearse, the model erroneously interpreted it as My Arms Are Your Grave. This hallucinatory output led to a low evaluation score, highlighting the model's vulnerability to semantic misinterpretations. The right case is a paralinguistic-featured utterance, where two sentences with identical text convey distinct meanings due to variations in prosodic pauses during human-recorded audio. The first sentence denotes that the musician is performing on stage for the fans, while the second suggests that the performance is specifically for the fans located on the stage. Notably, current models struggle to distinguish between such nuanced interpretations, indicating a limitation in handling paralinguistic cues.

## 4 RELATED WORK

Driven by recent advancements, the architecture of conversational models is shifting from traditional pipelines to end-to-end SpeechLLMs for real-time speech interaction (Zhang & Wang, 2019; Zhang et al., 2020; Jacqmin et al., 2022; Lee et al., 2021; Feng et al., 2023; Xie & Wu, 2024a; Fang et al., 2024; Défossez et al., 2024; Zhang et al., 2024; Song & Wang, 2024; Wang et al., 2024; Xie & Wu, 2024b; Long et al., 2025). However, the evaluation benchmarks remain underdeveloped and suffer from various issues, which in turn limit the progress of the field (Hassid et al., 2023; Mostafazadeh et al., 2016; Fang et al., 2024; Li et al., 2023; Yao et al., 2024).

Recently, researchers (Zeng et al., 2024; Yao et al., 2024) constructed evaluation sets from text benchmarks such as Web Questions (Berant et al., 2013) and TriviaQA (Joshi et al., 2017) to assess the model's knowledge question-answering ability. However, their evaluation relies on exact character matching, which leads to many false negatives. This problem was later alleviated by Baichuan-Audio's OpenAudio Bench (Li et al., 2025), which uses GPT-based scoring to determine answer correctness. Besides, benchmarks such as AIR-Bench (Perfetti et al., 2014) and SD-Eval (Ao et al., 2024) begin to explore more diverse acoustic conditions and speaker variability. However, neither benchmark is designed to support full S2S interaction or evaluate the expressive, generative capabilities of modern speech LLMs. We provide query examples in Appendix C for detailed comparison.

VoiceBench (Chen et al., 2024), one of the most popular speech chatbot benchmarks, converts commonly used text benchmarks (such as MMLU (Hendrycks et al., 2021) and OpenBookQA (Mihaylov et al., 2018)) into speech data through text-to-speech (TTS) synthesis to evaluate speech models. However, it has two main limitations. First, it directly uses text benchmark queries to evaluate speech models, but the query distribution for speech models differs significantly from that of text models. Second, the output modality used for evaluation remains text.

## 5 CONCLUSION

In this work, we addressed the pressing need for comprehensive and specialized benchmarks for end-to-end speech LLM evaluation. By systematically curating authentic user queries , introducing controlled variations in speaker attributes, and augmenting the dataset with realistic acoustic noise

and speech-specific challenges, we constructed a benchmark that rigorously tests model robustness in challenging acoustic environments. Furthermore, our development of a decoupled, dual-track evaluation framework enables a reliable assessment of both semantic correctness and speech quality.

## REPRODUCIBILITY STATEMENT

To facilitate reproducibility of our work, we provide detailed descriptions of the datasets in the main paper. Details of the human evaluation are available in Appendix B, while the evaluation prompts are listed in Appendix H and Appendix F. The source code is provided in the supplementary material.

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

## A   LARGE LANGUAGE MODEL (LLM) USAGE STATEMENT

In accordance with the conference policies on Large Language Model (LLM) usage, we hereby disclose the following: After completing the initial draft of this paper, we utilized an LLM to enhance grammar and polish the writing of this manuscript. No new research ideas, experimental designs, or scientific content were generated by the LLM. All scientific contributions, analyses, and conclusions presented in this work are solely those of the authors. We take full responsibility for the content of this paper, including all sections that have been revised or improved with LLM assistance. The LLM is not an author and did not contribute to the research ideation or substantive scientific writing.

This statement is provided to ensure transparency and compliance with the conference's policies on LLM usage.

## B   HUMAN EVALUATION DETAILS

We invited four annotators, each with over one year of experience in LLM evaluation, and two quality control (QC) specialists who have been trained for over two years. All our annotators and QC specialists are university graduates. The annotators performed the initial scoring, and the QC specialists reviewed all annotations for consistency and accuracy. The annotation process follows a two-stage protocol: (1) initial annotation by the four annotators, and (2) QC review. For ambiguous or complex cases, the final score was determined through discussion between the annotators and the QC group to ensure fairness and accuracy. The guidelines provided to annotators and QC specialists are identical to those used in our automatic evaluation, and prompts are provided in Appendix F.

## C   BENCHMARK QUERY COMPARISON

SD-Eval derives its queries from existing speech datasets. However, these datasets are not constructed from real user intents, and many of the queries, in fact, are declarative statements rather than actual queries (see Table 3). This limits their applicability to large model scenarios, as they do not represent meaningful, goal-directed requests to the model.

In contrast, AIR-Bench designs synthetic, text-based queries targeting audio inputs, while our benchmark is built on real user queries curated from WildChat. This approach better reflects authentic conversational goals and the linguistic diversity found in natural human interactions. To clearly demonstrate the differences, we have included a comparison table with representative cases.

## D   NATURAL NOISE DETAILS

To systematically evaluate the robustness of speech models in real-world environments, we augmented our benchmark's audio queries with a diverse set of non-linguistic natural noises. These noise samples were curated from high-quality, publicly available sound libraries to ensure realism and variety.

Table 4 provides a detailed breakdown of the five major categories of noise we selected: Animals, Natural soundscapes, Human non-speech sounds, Interior/domestic sounds, and Exterior/urban noises. The table also specifies the sampling ratio for each category, which represents its proportional representation in our final noise dataset, designed to reflect the commonality of these sounds in daily life. This structured approach to noise injection allows for a rigorous and fine-grained analysis of a model's ability to maintain performance under various challenging acoustic conditions.

## E   ASR FOR MCQ RESULTS

Evaluating the generated audio from end-to-end speech models presents a significant challenge, especially when attempting to use automated scoring methods. A common approach in text-based benchmarks is to use multiple-choice questions (MCQs) and check for an exact match. However, this method is unreliable for speech outputs when mediated by an Automatic Speech Recognition (ASR) system.

Table 3: Comparison of representative cases from SD-Eval, AIR-Bench, and WildSpeech-Bench, demonstrating the difference in query intent and complexity.

| Case | SD-Eval | AIR-Bench | WildSpeech-Bench (Ours) |
|---|---|---|---|
| Case 1 | **Query (audio):** Inside, the atmosphere was quiet. | **Context (audio):** I cannot, you are not here by choice. Nobody would ride this bus by choice. **Query (text):** What is the dominant emotion expressed in the speech? A. neutral B. angry C. sad D. happy | **Query (audio):** imagine that you are a UI design student and you need to develop ideas for unique applications that solve some of your personal problems. Ask me some questions to better understand the task and at the end give me several ideas for those apps. |
| Case 2 | **Query (audio):** Consumers eat other animals, and consumers can get eaten by other animals. | **Context (audio):** Stop it. Stop it. I hate you, do you hear? You're overbearing and conceited and utterly impossible. **Query (text):** What phrase is repeated by the speaker at the beginning of the speech? | **Query (audio):** What is the best way to invest as a 16 year old to make a million by 50? |

Table 4: Non-linguistic Noise Categories, Sampling Ratios, and Examples

| Category | Sampling Ratio | Examples |
|---|---|---|
| Animals | 0.10 | Dog, Rooster, Pig, Cow, Frog, Cat, Hen, Insects, Sheep, Crow |
| Natural soundscapes & water sounds | 0.20 | Rain, Sea waves, Crackling fire, Crickets, Chirping birds, Water drops, Wind, Pouring water, Toilet flush, Thunderstorm |
| Human, non-speech sounds | 0.20 | Crying baby, Sneezing, Clapping, Breathing, Coughing, Footsteps, Laughing, Brushing teeth, Snoring, Drinking sipping |
| Interior/domestic sounds | 0.25 | Door wood knock, Mouse click, Keyboard typing, Door wood creaks, Can opening, Washing machine, Vacuum cleaner, Clock alarm, Clock tick, Glass breaking |
| Exterior/urban noises | 0.25 | Helicopter, Chainsaw, Siren, Car horn, Engine, Train, Church bells, Airplane, Fireworks, Hand saw |

The primary issue is that the ASR system can introduce transcription errors, unfairly penalizing the speech model for the ASR's own mistakes. A model may generate a perfectly clear and correct spoken answer, but if the ASR misinterprets it, the response is marked as incorrect. This conflates the performance of the model being tested with the performance of the ASR evaluator.

Table 5 provides concrete examples of this failure mode. In Case 2 and Case 3, the speech model's likely correct spoken answers ("A. microorganisms" and "C. a town") are erroneously transcribed, leading to an incorrect evaluation. These examples highlight the unsuitability of relying solely on ASR for scoring spoken MCQ answers and underscore the need for more robust evaluation methodologies, such as our proposed framework, which are less susceptible to such cascading errors.

Table 5: Examples showing why multiple-choice questions are unsuitable for evaluating speech models.

| Case | Speech Output | ASR Transcription |
|------|---------------|-------------------|
| case1 | A. keeping warm | keeping warm |
| case2 | A. microorganisms | I'm microorganisms. |
| case3 | C. a town | see a town |

## F EVALUATION PROMPT

---

**ℹ Evaluation Prompt**

# Instructions

You are an evaluation expert. Your task is to assess the quality of AI model responses. We will provide you with user queries and AI responses. Please note that both the user queries and AI responses are in audio format. For your convenience, we have converted them into text, but you should evaluate from the perspective of voice communication and analyze the characteristics of voice communication when assessing the quality of the AI response.

You should first carefully read the user query to analyze the task, then evaluate the quality of the response based on the rules provided below.

# Conversation between User and AI

### User Query

```
<|begin_of_query|>
```

```
{query}
```

```
<|end_of_query|>
```

### AI Response

```
<|begin_of_response|>
```

```
{response}
```

```
<|end_of_response|>
```

# Evaluation

## Checklist

```
<|begin_of_checklist|>
```

```
{checklist}
```

```
<|end_of_checklist|>
```

The checklist serves as a guiding framework for your evaluation. However, feel free to consider aspects beyond its contents to ensure a well - rounded assessment.

## Rules

You should evaluate based on the analysis of user questions and AI responses, referring to the contents in the checklist during the evaluation. However, remember that the checklist is meant to provide comprehensive reference information, but it is not the standard answer. Sometimes, the AI

---

response does not need to cover all the contents involved in the checklist to meet user needs, and you need to make this judgment on your own. The scoring scale ranges from 1 to 10:
- 1~2 points: No value/meaningless. The AI response contains many factual errors or serious flaws, or is irrelevant to the user query, providing little to no value to the user.
- 3~4 points: Partially valuable/meaningful. The AI response contains several factual errors or serious flaws, or poorly meets the user's requirements, but has some redeeming qualities and offers partial value to the user.
- 5~6 points: Flawed. The AI response has some issues, such as minor factual errors/flaws, or does not fully meet the user's requirements. However, these are relatively minor, and the response generally satisfies the user's needs.
- 7~8 points: Meets requirements. The AI response satisfies the user's needs well, with no major flaws or errors, or only very minor issues that do not affect overall quality.
- 9~10 points: High quality. The AI response perfectly meets the user's requirements, with virtually no room for improvement.

## Output Format

First, analyze the query itself and understand the user's intent. Then provide your analysis of the model's response. Summarize your evaluation in two aspects: "Strengths" and "Weaknesses". Finally, write your score. The score should appear on the last line in the following format: Score: [your score]

# G  PERFORMANCE DEGRADATION IN DUAL-MODAL GENERATION

A notable advancement in modern Speech Large Language Models (SpeechLLMs) is the flexibility of their output generation. Many models can be configured to respond in a text-only mode or a dual-modal mode, which produces synchronized speech and text. This flexibility, however, can introduce performance trade-offs.

While Kimi-Audio demonstrates strong performance on speech-to-text dialogue tasks such as VoiceBench (Chen et al., 2024), we observed a significant performance drop on our speech-to-speech benchmarks. To investigate the cause of this discrepancy, we conducted a targeted evaluation of its different output modes on the CommonEval subset of VoiceBench . The results reveal a clear degradation contingent on the output mode. Specifically, the model achieved a high score of 3.97 in its text-only mode. However, when switched to the dual-modal setting, the score of the textual component dropped to 3.00. This indicates that the performance bottleneck is not only in the speech synthesis but also affects the quality of the underlying text generation when producing multi-modal outputs. We provide illustrative examples in Table 6.

# H  EVALUATION CHECKLIST

Standard evaluation of generative models, particularly when using another LLM as a judge, often suffers from issues of subjectivity and inconsistency. A generic prompt like "Is this a good response?" can lead to ratings that are overly influenced by the fluency or style of the output, rather than its factual accuracy or adherence to the query's core constraints. To address these challenges and enhance the reliability of our assessment, we introduce a query-specific **Evaluation Checklist** for each item in our benchmark.

This checklist serves as a detailed, structured rubric that guides the evaluation process. Instead of a single holistic judgment, the evaluator is tasked with verifying a set of pre-defined, critical criteria tailored to the unique demands of each query. As illustrated in Table 7, these checklists serve multiple functions:

- **Ensuring Factual Accuracy:** For information-inquiry tasks (e.g., the "bradypnea" query), the checklist provides key facts that a correct response must contain, transforming the evaluation from a plausibility check into a verification of correctness.
- **Mandating Structural Integrity:** For complex requests (e.g., troubleshooting Chrome), the checklist defines the expected structure of a helpful answer, such as separating causes from solutions.

Table 6: Comparison of Model Outputs Across Different Generation Modes.

| Case | Prompt | Mode | Response | Score |
|---|---|---|---|---|
| 1 | What do you think about songs sung in unison? | Dual-Mode (Text) | It sounds like you're interested in singing together! | 2 |
| | | Text-Mode | What a beautiful topic!! Singing songs in unison can be a truly magical experience! It's a powerful way to connect with others, create a sense of community... (excerpt) | 5 |
| 2 | Can I use Last.fm to listen to Twilight Time by Dj Khalil? | Dual-Mode (Text) | It sounds like you're looking for a way to listen to "Twilight Time" by DJ Khaled. I'm here to help! | 2 |
| | | Text-Mode | Yes, you can use Last.fm to listen to "Twilight Time" by DJ Khaled. Last.fm is a music streaming service that allows you to listen to music online... (excerpt) | 3 |
| 3 | I wonder if plastic plants make a room more beautiful or more melancholic. | Dual-Mode (Text) | Plastic plants can add a touch of greenery to a room, but they might not evoke the same emotional response as real plants. What do you think? | 4 |
| | | Text-Mode | What an intriguing question!! The impact of plastic plants on a room's aesthetic and emotional atmosphere can vary depending on several factors... (excerpt) | 4 |

- **Focusing on Critical Nuances:** For queries testing deep understanding (e.g., the "prosodical" example), the checklist explicitly directs the evaluator's attention to the most crucial element—such as the semantic shift caused by punctuation—and defines the failure to capture it as a serious error.

By anchoring the evaluation to these concrete and query-aware criteria, our method significantly reduces ambiguity and ensures that the final scores reflect a more accurate, consistent, and fine-grained measure of the model's true capabilities.

## I  LIMITATIONS AND FUTURE WORK

While the Query-Aware Checklist introduced in this work significantly enhances evaluation consistency, we acknowledge a potential long-term risk. The widespread adoption of any standardized benchmark creates an incentive for developers to optimize systems specifically for its criteria. This phenomenon, often described as "teaching to the test," could shift development focus from improving general conversational abilities to explicitly matching patterns in the checklist. Consequently, a model's score on WildSpeech-Bench might improve without a proportional advancement in real-world performance. Addressing this potential negative externality represents a critical area for future research within the AI benchmarking community.

Table 7: Query Evaluation Checklist

| Query | Evaluation Checklist | Type |
|-------|---------------------|------|
| Describe the optimal gear ratio for fixed gear cycling | • Examples of the best gear ratio for a fixed gear bike:
  – Mountain bike: Use 32:36, 34:32 and other low tooth ratio combination.
  – Commuter and city bikes: Lower gear ratios (e.g. 40:16 or 44:16) are generally easier to step on in traffic and all kinds of terrain.
  – Racing and flat terrain: Higher gear ratios (e.g., 48:13 or 50:14) are often used for faster speeds on flat or slightly undulating terrain.
• Pay attention to whether there is a calculation error in the gear inches in the AI response. If there is a problem of mathematical calculation error, it is regarded as a serious defect. | Information-Inquiry |
| i opened google chrome yesterday, and i noticed that suddenly all my customized settings were gone. that includes the theme and the pinned extensions. what's happened and is there any fix? | • The sudden disappearance of custom settings in Google Chrome may be due to the following reasons and solutions:
  – **Reasons:** Software conflict, Browser cache issue, Impact of system updates, Corrupted configuration file or synchronization error, Incorrect operation.
  – **Solutions:** Clear the browser cache, Reset the browser settings, Check and modify the shortcut, Reinstall Google Chrome. | Request For Solution |
| Please tell me what the following two sentences mean: A woman, without her man, is nothing./A woman: without her, man is nothing. | • First sentence: A woman without her man is nothing; Second sentence: A man without women is nothing.
• Note that the meanings of these two sentences are completely opposite, emphasizing the importance of men and women respectively. If this problem is not found, it indicates a serious error in the understanding of the sentence meaning. | Prosodical |

## J    BENCHMARK DESIGN RATIONALE

### J.1    FOCUS ON SINGLE-TURN EVALUATION

The design of WildSpeech-Bench is deliberately focused on the single-turn evaluation setting. This decision is rooted in the principle that a **meaningful multi-turn evaluation cannot be built upon flawed single-turn foundations**. The core contribution of our work is to establish this solid foundation.

Prior to WildSpeech-Bench, existing benchmarks lacked a systematic approach to addressing fundamental challenges within a single conversational turn, such as direct speech quality assessment, robustness to acoustic noise, and query-aware relevance. Attempting to construct a multi-turn benchmark without first resolving these core issues would only compound the evaluation problem, leading to unreliable and confounded results. Our work provides the necessary tools to first get a reliable signal on single-turn performance, paving the way for future, more robust multi-turn benchmarks.

### J.2    DATASET SCALE

While the total number of queries is focused, each of the 1,100 queries was manually reviewed and validated to ensure it meets our standards for commonality, difficulty, and suitability for spoken interaction. These queries are then paired with a custom-designed checklist to enable query-aware evaluation, significantly increasing the annotation and verification workload. This rigorous process ensures high per-sample evaluation quality and diagnostic value. We also note that several widely adopted benchmarks follow a similar **high-quality, focused-scale design philosophy**; for example, AlpacaEval (Li et al., 2023) contains only 805 examples, the original MT-Bench (Zheng et al., 2023) comprises 80 queries, and WildBench (Lin et al., 2024) contains 1024 queries . These benchmarks are valued not for their size, but for their precision and carefully controlled evaluation. To the best of our knowledge, WildSpeech-Bench is the first speech benchmark to offer such a query-aware evaluation mechanism, prioritizing diagnostic depth over sheer volume."

