# OpenReview forum: "WildSpeech-Bench: Benchmarking End-to-End SpeechLLMs in the Wild"
_ICLR.cc/2026/Conference — Submitted to ICLR 2026_

### Official Review · Reviewer_PXjr · 2025-10-29

**Soundness:** 2
**Presentation:** 2
**Contribution:** 2
**Rating:** 4
**Confidence:** 4

**Summary:**

The paper introduces WildSpeech-Bench, a new benchmark for evaluating end-to-end (E2E) Speech-to-Speech (S2S) Large Language Models (SpeechLLMs). The authors claim that existing benchmarks are ill-suited for this task, as they are often adapted from text-based benchmarks, evaluate Speech-to-Text (S2T) output, and use unrealistic, clean, synthesized audio queries .

The paper's primary contributions are:

- A collection of 1,100 queries. This set is composed of 1,000 "general" queries sourced from the WildChat dataset (real-world user-ChatGPT text interactions) and 100 "Paralinguistic-Featured" (PF) queries that were manually crafted and recorded by humans to test nuances like pause, stress, and tone . The audio for all queries is augmented with "Human Noise" and "Natural Noise" to simulate real-world conditions .

- The paper proposes an automatic evaluation method where an LLM (GPT-4o-mini) scores the model's transcribed response. To improve accuracy, this LLM-judge is guided by "meticulously hand-crafted checklists" specific to each query, which are designed to assess factual, structural, and semantic criteria .

**Strengths:**

- The concept of a manually crafted, human-recorded set of queries to test nuanced paralinguistic understanding is new.
- Using per-query, hand-crafted checklists to guide LLM-judges is a clever way to improve reliability and reduce the ambiguity of generic prompts.
- The systematic inclusion of both human and environmental background noise is a good feature, pushing models beyond idealized "clean room" performance .

**Weaknesses:**

- The benchmark's core claim is evaluating SpeechLLMs "in the wild". This claim is not met for two fundamental reasons:
  - The benchmark is 100% focused on single-turn evaluation. Real-world, "in-the-wild" speech conversations are almost never single-turn. They are interactive and multi-turn. This benchmark completely fails to evaluate core conversational abilities such as context maintenance, coreference resolution, or managing a coherent dialogue over time.
  - 90% of the benchmark (1,000 out of 1,100 queries) is not human-recorded but synthesized using voice-cloned TTS. While diverse profiles are used, this is still synthesized speech, not authentic human speech "in the wild." The only truly "wild" acoustic data is the 100-item PF set.

- The dataset contains only 1,100 queries in total. This scale is insufficient for a robust and generalizable benchmark. The authors' justification (comparing it to MT-Bench's 80 queries) is not persuasive. Evaluating the vast, multi-faceted domain of S2S interaction (including acoustic robustness, paralinguistics, and content quality) requires a much larger sample. This small, fixed set is highly susceptible to "teaching to the test," as the authors themselves acknowledge . Models can easily be fine-tuned to perform well on this specific set of 1,100 queries and their corresponding checklists, rendering the benchmark obsolete quickly.

- The paper's evaluation methodology contains a contradiction.
  - It criticizes prior benchmarks for "focus on evaluating the text output rather than the generated speech".
  - However, its own primary evaluation framework does exactly that. The model's audio response is transcribed by Whisper-large-v3, and this text transcription is then scored by the GPT-4o-mini judge .
  - This is, by definition, a Speech-to-Text (S2T) evaluation, albeit a more complex one using an LLM-judge. The paper fails to explain how an LLM reading text can assess "speech-centric criteria like clarity... and tonal appropriateness". The paper even dedicates an entire appendix (Appendix E) to demonstrating how ASR errors make evaluation unreliable , yet it builds its entire scoring system on top of an ASR system.

**Questions:**

- How could you claim this benchmark evaluates models "in the wild" when it contains zero multi-turn conversations, which are the defining characteristic of "in-the-wild" interactions?
- The paper's central criticism of other benchmarks is their reliance on evaluating text output. How do you justify that your own primary evaluation method—which transcribes speech to text and has an LLM judge that text —is not guilty of the exact same flaw?
- How does your text-based LLM-judge assess "speech-centric criteria" like "tonal appropriateness"? Are you claiming the ASR transcription captures this, or is this specific criterion simply not evaluated by the LLM-judge?
- Given the small, fixed scale of 1,100 queries and their hand-crafted checklists, what prevents model developers from simply fine-tuning their models on your benchmark data, leading to inflated scores that do not reflect true general-purpose S2S capability?

---

> ### Author Response · Authors · 2025-11-21
> **Author Response to Reviewer PXjr (Part 1/2)**
>
> We thank Reviewer PXjr for the insightful feedback. While we agree that S2S evaluation presents complex challenges, we would like to clarify **a central misunderstanding** regarding our approach. Below, we address the points raised in the order they were presented.
>
> ## On the "In the Wild" Claim (Weakness 1 & Question 1)
> We will address the reviewer's two points:
>   - **On the Single-Turn Focus:** This focus was a deliberate design choice. We align with current major benchmarks for SpeechLLMs (e.g., VoiceBench [1], AIR-Bench [2], MMAU [3]), which are also primarily single-turn. We argue that the field must first solve the prerequisite challenge of single-turn robustness—specifically regarding noise and paralinguistics—before advancing to complex dialogues, as **meaningful multi-turn evaluation cannot be built upon flawed single-turn foundations**.
>   - **On the TTS/Authenticity:** The reviewer's core concern is that 90% synthetic speech invalidates our "in-the-wild" claim. The **VoiceBench [1]** paper rigorously studied this exact issue. They found that model rankings remain **"consistent"** and show a **"strong correlation"** when evaluated on synthetic *vs.* real human speech. This strongly indicates that using high-quality, diverse TTS (we used **16 distinct voice profiles** ) is a **valid and reliable methodology** for benchmarking.
>   - Consequently, our "in-the-wild" claim is defined by critical complexity factors that **previous work failed to adequately address**:
>     - **Real Queries** from actual user logs (WildChat) ;
>     - **Real Acoustic Noise**, which we systematically added to all samples ;
>     -  **Real Human Recordings for Challenging Queries** (100% of the PF set), precisely where TTS is known to fail (paralinguistics) .
>
> ## On Dataset Scale  (Weakness 2 )
> We address the concern regarding dataset scale from two critical perspectives: **Practical Necessity** and **Statistical Validity**.
>
> #### **A. Practical Necessity: The S2S Evaluation Bottleneck**
>
> We must emphasize that S2S evaluation faces unique challenges fundamentally different from T2T evaluation: **the severe cost and slowness of the evaluation pipeline itself.** A massive benchmark would be computationally prohibitive for the community due to the four-stage pipeline required for *each single sample*:
> 1. **Slow S2S Inference:** Generating audio involves complex architectures (e.g., GLM-4-Voice's interleaved tokens).
> 2. **Slow Vocoder Step:** A separate, computationally intensive vocoder is required to render the waveform.
> 3. **Slow ASR Pass:** The audio must undergo a full ASR transcription (e.g., Whisper-large-v3) for Track 1 (Content) evaluation.
> 4. **Slow Acoustic Pass:** The audio is processed *again* by an acoustic metric (e.g., UTMOS) for Track 2 (Speech) evaluation.
>
> Given these bottlenecks, we made a deliberate trade-off: we **"prioritized diagnostic depth over sheer volume"**, investing in manual curation and hand-crafted checklists.
>
>
> #### **B. Statistical Validity: Proving Robustness**
>
> The reviewer argues that this scale is "insufficient for a robust... benchmark." To empirically refute this and prove that our 1,100 samples provide sufficient statistical power to distinguish between models, we **conducted pairwise t-tests** on all model comparisons.
> Appendix Table X: Pairwise p-values (one-tailed) for Overall Avg. Scores (N=1100)
> (P-value for H_a: Row Model Score > Column Model Score)
> | | Naive Pipeline (5.24) | Kimi-Audio (4.54) | GLM-4-Voice (5.03) | MiniCPM (5.08) | Qwen-2.5-omni (6.01) | GPT-4o-Audio (6.29) |
> | :--- | :---: | :---: | :---: | :---: | :---: | :---: |
> | **Naive Pipeline (5.24)** | - | **<0.01*** | **<0.01*** | **<0.01*** | p>0.99 | p>0.99 |
> | **Kimi-Audio (4.54)** | p>0.99 | - | p>0.99 | p>0.99 | p>0.99 | p>0.99 |
> | **GLM-4-Voice (5.03)** | p>0.99 | **<0.01*** | - | p=0.83 | p>0.99 | p>0.99 |
> | **MiniCPM (5.08)** | p>0.99 | **<0.01*** | p=0.17 | - | p>0.99 | p>0.99 |
> | **Qwen-2.5-omni (6.01)**| **<0.01*** | **<0.01*** | **<0.01*** | **<0.01*** | - | p>0.99 |
> | **GPT-4o-Audio (6.29)** | **<0.01*** | **<0.01*** | **<0.01*** | **<0.01*** | **<0.01*** | - |
> **Analysis:**
> 1. **Clear Separation of Tiers:** As the table shows, the performance gaps between major tiers (e.g., GPT-4o vs. Qwen, or Qwen vs. Naive Pipeline) are statistically significant (p < 0.01). This proves that 1,100 samples are more than sufficient to robustly rank these models.
> 2. **Fidelity:** The test also correctly identifies when models are statistically tied (e.g., GLM-4-Voice vs. MiniCPM, p=0.17/0.83), confirming the benchmark's fidelity in recognizing similar performance.
>
> In conclusion, while "more data" is always desirable, our analysis confirms that the current scale achieves **high statistical robustness** while remaining practically usable for the research community.
>
> **Due to character limitations, our response continues in the following comment.**

---

> > ### Comment · Reviewer_PXjr · 2025-11-27
> >
> > Authors in their response have confirmed that the benchmark is restricted to single-turn interactions and relies 90% on synthesized (TTS) audio. In my opinion it fundamentally contradicts the paper's main claim of "in the wild" evaluation and highlights the gap between paper's main claim and its actual contributions. I'll maintain my score.

---

> > > ### Author Response · Authors · 2025-11-27
> > > **Clarification on the Scope of "In the Wild" and Scientific Validity**
> > >
> > > We thank the reviewer for the continued engagement. We understand your reservation regarding the strict definition of "in the wild." It appears our disagreement stems from **differing interpretations of this term**, rather than the utility or validity of the benchmark itself.
> > >
> > >  ## **Reconciling the "In the Wild" Definition**
> > >
> > > **Your Definition**: Focuses on Conversational Dynamics (Multi-turn) and Source Origin (100% Human Recording). We agree this is the ultimate "north star" for the field.
> > >
> > > **Our Contribution**: Focuses on Acoustic Realism (Real-world Noise/Interference) and Intent Realism (Real User Queries from WildChat).
> > >
> > > Why our contribution matters: Current SOTA models (like GPT-4o-audio) fail significantly when exposed to the "Acoustic Wild" (noise) and "Intent Wild" (diverse/paralinguistic queries), as our results prove. **Even in single-turn interactions, models are currently fragile**. Our benchmark provides the first systematic tool to measure and improve this specific, critical robustness.
> > >
> > > ## **The Scientific Validity of the Hybrid Approach**
> > > You expressed concern that reliance on synthesized audio (TTS) contradicts the "in the wild" claim and invalidates the results. We respectfully remind the reviewer of the **scientific evidence** presented in our rebuttal, which directly supports the validity of our design choice:
> > >
> > > Consistency of Trends (External Evidence): As noted in our previous response, concurrent research (VoiceBench, Chen et al., 2410.17196v3) rigorously studied this issue. They found that **model rankings remain "consistent" and show a "strong correlation"** when evaluated on synthetic vs. real human speech. This confirms that high-quality TTS is a valid scientific proxy for measuring capability.
> > >
> > > Statistical Confidence (Internal Evidence): Furthermore, our paired t-tests confirm that the performance gaps we report are **statistically significant (p < 0.05)**.
> > >
> > >
> > > We hope you agree that a benchmark exposing significant robustness failures in SOTA models (e.g., -2.10 drop in S2S vs S2T) has high merit for the community, regardless of whether it meets the strictest possible definition of "wild."
> > >
> > > We sincerely hope this clarification addresses the core of your objection. We believe discarding these findings due to a definitional difference would be a loss for the community's effort to build more robust SpeechLLMs.
> > >
> > > ## **Prioritizing Scientific Insight over Definitional Differences**
> > > We respectfully argue that a benchmark exposing significant robustness failures in SOTA models (e.g., the -2.10 drop in S2S vs S2T performance) holds high merit for the community, regardless of whether it meets the strictest possible definition of "wild."
> > >
> > > To be candid, your review was the **most challenging one we received**, but your push for rigorous validation was **particularly instrumental in strengthening the paper's core arguments**. While we may still hold slightly different views on the strict terminology, we hope you can see that the substance of this work has been elevated by your feedback. We earnestly hope that you will consider these substantial improvements and the empirical value of our findings in your final assessment.

---

> > > > ### Comment · Reviewer_PXjr · 2025-11-27
> > > >
> > > > I have carefully considered the authors' rebuttal, specifically their argument that prior studies have established a precedent for single-turn, synthetic benchmarks in this domain.
> > > >
> > > > However, my assessment as a reviewer is isolated to the specific claims and contributions presented in this manuscript. The central claim of this paper—prominently featured in the title and introduction—is that it evaluates models "in the wild." In the broader machine learning and speech communities, this term carries a specific semantic weight: it implies natural, uncurated, human distributions, which inherently include multi-turn dynamics and authentic human prosody.
> > > >
> > > > The authors' own methodology contradicts this claim. On Line 170, the paper explicitly states: "After removing multi-turn queries...". This confirms that the authors actively filtered out the defining characteristic of "wild" conversational data. By removing dialogue dynamics and relying on 90% synthetic audio, the benchmark essentially evaluates models "in the noise" (due to added background sounds) rather than "in the wild."
> > > >
> > > > Furthermore, I find the ASR-based judge insufficient for speech-to-speech benchmark. A high correlation on a limited set of 1,100 queries might just mean that for these specific synthetic queries, the ASR is "good enough" at capturing content. It does not prove that an ASR-based judge can capture the paralinguistic nuances (prosody, emotion, timing) that are the entire point of a speech-to-speech benchmark. If the ASR misses the tone but gets the text right, and the model gets a high score, the benchmark has failed to measure true speech-to-speech capabilities—even if the score aligns with a human judge who focused primarily on content.
> > > >
> > > > I recognize the value of this work as a robustness benchmark for noisy, single-turn interactions. However, labeling it "in the wild" is scientifically misleading.
> > > >
> > > > If the authors are willing to drop the misleading "in the wild" branding (e.g., renaming the paper/benchmark to something accurate like "RobustSpeech-Bench" or "Noise-Augmented Evaluation" and clarifying these changes in the introduction and remaining paper), I am open to updating my score to reflect the paper's true contribution.
> > > >
> > > > If the current branding remains, I cannot recommend acceptance, as the claims do not match the artifact. For now, I maintain my rating.

---

> > > > > ### Author Response · Authors · 2025-11-27
> > > > >
> > > > > We sincerely thank the reviewer for the candid feedback. We **fully accept** your critique and are fully committed to correcting this misalignment. We have taken the following **immediate actions**:
> > > > >
> > > > > 1. Total Removal of "In the Wild" Claims : We have performed a "clean sweep" of the abstract and introduction, **removing every instance where the phrase "in the wild" was used** to describe the audio data. We now describe the data precisely as "WildChat-derived queries augmented with realistic noise" .
> > > > >
> > > > > 2. Title Revision: To ensure no reader is misled about the audio source or conversational dynamics, we have **revised the paper title as you suggested to remove the "in the wild" claim** and replace it with a precise technical description:
> > > > >     - Old: "WildSpeech-Bench: Benchmarking End-to-End SpeechLLMs in the Wild"
> > > > >     - New: "WildSpeech-Bench: Benchmarking End-to-End SpeechLLMs on Noise-Augmented Real-World Queries"
> > > > >
> > > > > 3. Revised PDF: We have **already updated the PDF** to reflect these rigorous de-scoping changes.
> > > > >
> > > > > 4. Justification for the Identifier "WildSpeech" (Data Lineage): Regarding the specific identifier "WildSpeech," we respectfully propose retaining this name strictly to **denote its data lineage**: the queries are derived from the "WildChat" dataset . It indicates "Speech synthesized from WildChat data" (Real-World Queries), ensuring the diversity of user intent matches real-world distributions, rather than implying "wild" acoustic recording environments.
> > > > >
> > > > >
> > > > > We earnestly hope that you will consider these substantial improvements and the empirical value of our findings in your final assessment.

---

> > > > > ### Author Response · Authors · 2025-11-28
> > > > > **Verification of PDF Update and Renaming**
> > > > >
> > > > > We are writing to respectfully confirm that the **revised PDF** has been successfully uploaded. As promised in our previous response, this revision incorporates the major changes you suggested:
> > > > >
> > > > > - Renaming: The title is now "WildSpeech-Bench: Benchmarking End-to-End SpeechLLMs on Noise-Augmented Real-World Queries."
> > > > >
> > > > > - De-scoping: We have removed the "in the wild" claims from the abstract and introduction to align strictly with the technical artifacts.
> > > > >
> > > > > We hope these decisive actions fully resolve your concerns regarding the benchmark's positioning. We remain available for any further clarifications.

---

> ### Author Response · Authors · 2025-11-21
> **Author Response to Reviewer PXjr (Part 2/2)**
>
> Continuing from the previous comment, here we provide our responses to the rest questions.
>
> ## On Evaluation Framework（Weakness 3 & Questions 2、3 ）
> We appreciate the reviewer's scrutiny regarding our evaluation methodology. We clarify that there is **no contradiction**; rather, our design is a deliberate response to the limitations of current tools. The reviewer’s concern stems from conflating the **"modality being evaluated"** with the **"means of evaluation."**
>
> ### 1.The Logic Chain: From Limitation to Design
> The reviewer correctly notes our analysis of ASR errors (Appendix E). Far from ignoring this, Appendix E is the very evidence that necessitates our novel design. Our logic flows as follows:
>
> (1) **The Gap**: A model's Speech-mode output is not semantically identical to its Text-mode output (proven by the -2.10 performance drop in the table below).
>
> $\downarrow$
>
> (2) **The Necessity**: Therefore, we must evaluate the generated speech directly (S2S), rather than relying on text-only benchmarks (S2T).
>
> $\downarrow$
>
> (3) **The Constraint**: However, evaluating speech semantics requires ASR, but **ASR errors make rigid formats (like MCQs)** unreliable (as shown in Appendix E).
>
> $\downarrow$
>
> (4) **The Solution (WildSpeech)**: We specifically design Open-ended Queries (which are **robust to** minor ASR errors) and decouple the evaluation into two parallel tracks.
>
>   -  **Semantic Evaluation (Track 1):** We use ASR as a *tool* to extract content for semantic scoring. To ensure robustness against the transcription errors noted in Appendix E, we deliberately avoided brittle formats like MCQs and used open-ended queries. Critically, we empirically validated this approach against **human experts who evaluated the raw audio directly**. The high correlation (Pearson r=0.86) between our ASR-based scores and human scores confirms that our pipeline effectively mitigates transcription errors and aligns with human perception .
>   -   **Audio Quality Evaluation (Track 2):** We assess acoustic and paralinguistic quality directly on the generated audio, completely bypassing ASR. This is achieved through UTMOS scores, ensuring that speech-centric qualities are evaluated in their native modality.
>
>  **Table, S2T vs S2S performance (new experiment)**
> | Metric | Overall Average Score | TC (Text Creation) | II (Information Inquiry) | SR (Solution Request) | OE (Opinion Exchange) | PF (Paralinguistic) Avg. |
> | :--- | :---: | :---: | :---: | :---: | :---: | :---: |
> | S2T | 6.64 | 6.80 | 6.49 | 7.09 | 6.47 | 5.41 |
> | S2S  | 4.54 | 4.45 | 4.33 | 4.79 | 4.70 | 4.92 |
> | Difference | -2.10 | -2.35 | -2.16 | -2.30 | -1.77 | -0.49 |
>
> ### 2.On Assessment of "tonal appropriateness"
> We clarify that our text-based LLM-judge **does not** assess speech-centric criteria like "tonal appropriateness." We emphatically agree with your premise that ASR transcriptions cannot capture these nuances.Furthermore, regarding the measurement of "tonal appropriateness":
> 1. **Nature of Tasks:** We deliberately excluded "contextual tonal appropriateness" (e.g., matching an emotion to a situation) as a primary metric because the vast majority of our benchmark queries are **objective and functional** (e.g., *Information Inquiry*, *Solution Request*).
> 2. **Subjectivity of "Correct" Tone:** For these objective tasks, defining a single "correct" tone is highly subjective and ill-defined. For example, when answering a factual question, a "professional/steady" tone and a "cheerful/friendly" tone are often equally valid.
> 3. **Benchmark Scope:** Therefore, our benchmark prioritizes **semantic correctness** and **speech naturalness and quality** (as measured by UTMOS), rather than the model's "acting" capability or anthropomorphic emotional alignment, which is difficult to quantify objectively for functional queries.
>
>
> ## On the risk of data leakage（Question 4）
>  We must emphasize that this risk is **independent of the benchmark size**. Even if the dataset were expanded 10x, a malicious developer could still include it in their training data. This challenge—preventing models from overfitting a fixed test set—requires advanced **test set detection** tooling and community vigilance.
>
>
> ## Reference
>   1.   Chen, Yiming, et al. "Voicebench: Benchmarking llm-based voice assistants."
>   2.   Yang, Qian, et al. "Air-bench: Benchmarking large audio-language models via generative comprehension."
>   3.   Sakshi, S., et al. "Mmau: A massive multi-task audio understanding and reasoning benchmark."
>
>
> We are prepared to offer immediate clarification on any remaining points. We sincerely thank you for this rigorous critique, which has been **instrumental** in refining the paper's narrative and reinforcing its core contribution. We would appreciate your reconsideration of the paper's contribution.

---

### Official Review · Reviewer_rPtm · 2025-10-30

**Soundness:** 3
**Presentation:** 3
**Contribution:** 3
**Rating:** 6
**Confidence:** 3

**Summary:**

This paper argues that existing evaluation methods for large speech language models often rely on text-based benchmarks, thereby overlooking the unique characteristics and challenges inherent to spoken language—such as prosody, homophones, stuttering, and divergent user expectations. To address this gap, they introduce the first systematic benchmark specifically designed to evaluate speech-to-speech dialogue systems in realistic conversational scenarios. They detail the full pipeline for constructing this benchmark and propose a novel approach that leverages tailored evaluation checklists to enhance the accuracy of automated assessments. Experimental results demonstrate that the automated evaluation correlates well with human judgments, and further conduct ablation studies to validate the effectiveness of our strategy for injecting noise into user queries.

**Strengths:**

1. The authors present the first comprehensive benchmark designed to systematically evaluate end-to-end SpeechLLMs in practical speech-based conversations, along with an effective pipeline for constructing such benchmarks.
2. The authors propose a query-aware evaluation method that leverages customized evaluation checklists and prompts to improve the accuracy of automatic evaluation, and demonstrate that this approach aligns more closely with human judgments.

**Weaknesses:**

1. During evaluation, the authors adopted a checklist-based approach to provide the large language model with comprehensive reference information and then asked it to score model responses based on this information. If that’s the case, why not explicitly check each item in the checklist individually and assign scores based on whether each criterion is met? This may yield more precise and reliable evaluation results.

2. The current categorization of scenarios in the dataset is not yet sufficiently clear. Providing more explicit and well-defined scenario categories would help clarify the intended application scope and strengths of this benchmark.

3. In the naive pipeline, it's great to include additional results where the ASR model is replaced with a captioning model for comparison.

**Questions:**

Please refer to weaknesses.

---

> ### Author Response · Authors · 2025-11-21
> **Author Response to Reviewer rPtm**
>
> We thank Reviewer rPtm for the insightful comments on our methodology.
> ## **On Weakness 1: "Why not check checklist items directly?"**
> This is an excellent question. We use the checklist as a "guiding framework" for a 1-10 score, rather than a binary pass/fail, to capture **granularity**.
> A binary check struggles to score responses that are *partially* correct or conceptually right but miss a nuance (e.g., a response that gives 3 of 4 requested steps). Our 1-10 scale (e.g., "5-6 points: Flawed" vs. "7-8 points: Meets requirements" ) allows for this finer-grained assessment. However, we agree your proposed method is a valuable alternative, and we will add this as a discussion point for future work .
> ## **On Weakness 2: "Clarity of scenario categories"**
> We apologize if our categories were unclear. We define the five categories (TC, II, SR, OE, PF) in **Section 2.1.1** and provide explicit definitions and concrete examples for each in **Figure 2** . We will add a stronger reference to Figure 2 in the main text.
> | Category | Definition/Scope |
> | :--- | :--- |
> | **Information Inquiry (II)** | Users search for and obtain relevant information from various sources. |
> | **Solution Request (SR)** | Users encounter problems and seek specific action plans or executable plans. |
> | **Opinion Exchange (OE)** | Users ask for the AI's opinions for some relatively subjective questions.  |
> | **Text Creation (TC)** |  Text-related tasks such as creating stories, poems, etc.  |
> | **Paralinguistic-featured Utterance (PF)** | User-specific categories in voice conversation scenarios. |
>
> To prove these categories are not just labels but are **methodologically distinct and necessary**, we have **conducted a new analysis** of model rankings *within* each category (based on data from **Table 2**). As shown in the table below, model ranks change significantly, proving they test different capabilities. For example, **Qwen-2.5-omni** demonstrates a strong capacity for step-by-step reasoning by achieving **Rank 1** in *Solution Request (SR)*, but its rank falls sharply to **Rank 6** in the highly nuanced *Paralinguistic-Featured (PF)* category. Conversely, **MiniCPM** shows a more balanced profile across generative tasks, achieving **Rank 3** in *Opinion Exchange (OE)* but dropping to **Rank 5** in *Solution Request (SR)*, revealing distinct strengths in its underlying architecture.
>
> | Model | TC Rank | II Rank | SR Rank | OE Rank | PF Rank |
> | :--- | :---: | :---: | :---: | :---: | :---: |
> | **GPT-4o-Audio** | 1 | 1 | 2 | 1 | **1** |
> | **Qwen-2.5-omni** | 2 | 2 | **1** | 2 | **6** |
> | **Naive Pipeline** | 3 | 3 | 3 | 4 | 3 |
> | **MiniCPM** | 4 | 4 | 5 | 3 | 4 |
> | **GLM-4-Voice** | 5 | 5 | 4 | 5 | 5 |
> | **Kimi-Audio** | 6 | 6 | 6 | 6 | 2 |
>
> ##  **On Weakness 3: "Adding a captioning model"**
> This is an excellent suggestion. We have **conducted this new ablation study** to investigate the trade-offs between transcription and description in the speech-to-text front-end.
> We compared our original **Naive Pipeline (ASR)** (which uses Whisper-large-v3  for verbatim transcription) with a new "Naive Pipeline (Captioner)" baseline, utilizing the **Qwen3-Omni-Captioner** model. In the captioner pipeline, the LLM is explicitly prompted that the input is an **audio description** rather than a literal transcription.
> **(New Ablation Table: ASR vs. Captioner Pipeline)**
> | Pipeline Version | TC | II | SR | OE | PF | Overall Avg. |
> | :--- | :---: | :---: | :---: | :---: | :---: | :---: |
> | **Naive Pipeline (ASR)**  | 5.55 | 4.98 | 5.51 | 5.18 | 4.84 | **5.24** |
> | **Naive Pipeline (Captioner)** | 4.84 | 4.25 | 4.44 | 2.97 | 4.37 | 4.35 |
>
> As the table shows, the ASR-based pipeline **significantly outperforms** the Captioner-based pipeline across all categories (Overall Avg: 5.24 vs. 4.35).
>
> Our analysis suggests this is because captioning models tend to generate a significant amount of **semantically irrelevant information** (e.g., descriptions of timbre, speaker gender, background noise, or emotion) alongside the content. For the downstream LLM, this extraneous metadata acts as **noise or distraction**, interfering with its ability to focus on the precise semantic constraints and factual keywords required by the evaluation checklists. This validates our choice of a verbatim ASR-based pipeline as the stronger baseline for this benchmark.
>
>
> We remain fully available throughout the discussion period to address any further questions. We are eager to engage in continued dialogue to fully resolve your concerns

---

> > ### Comment · Reviewer_rPtm · 2025-11-27
> > **Response to Authors**
> >
> > Thanks for your response. I'm willing to maintain my current score.

---

> ### Author Response · Authors · 2025-11-27
>
> We sincerely thank you for your feedback.
>
> We are glad that our response addressed your concerns regarding the checklist evaluation method and the scenario categorization. In particular, we believe the **additional experiments comparing the ASR model with the captioning model (as per your suggestion)** have significantly strengthened the baseline analysis and provided clearer insights into the pipeline's robustness.
>
> Given that you recognized this work as the "first comprehensive benchmark" for end-to-end SpeechLLMs and validated the effectiveness of our proposed evaluation pipeline, we hope that the resolved concerns and the added comprehensive experiments serve to clarify the contribution and robustness of our work. If you agree that the paper has been strengthened, **we would be grateful if you might consider reflecting this in your final assessment**.

---

### Official Review · Reviewer_DAeh · 2025-10-31

**Soundness:** 3
**Presentation:** 3
**Contribution:** 3
**Rating:** 6
**Confidence:** 3

**Summary:**

This paper introduces WildSpeech-Bench, a novel benchmark designed for the end-to-end evaluation of Speech Large Language Models (SLLMs) in real-world, conversational scenarios. The authors identify significant gaps in existing benchmarks, such as an over-reliance on text-based queries, a lack of acoustic diversity, and evaluation methods that fail to capture the nuances of spoken language. To address this, WildSpeech-Bench is constructed by curating realistic queries from actual user-ChatGPT conversations (via the WildChat dataset) and generating speech data with diverse speaker attributes (age, gender, timbre) and background noise conditions. Speacially, authors proposed "Paralinguistic-Featured" queries to test a model's ability to handle speech-specific challenges like stuttering, prosodic ambiguity, and homophones. The benchmark also proposes an automatic evaluation protocol that uses query-aware checklists and LLM-based semantic scoring.

**Strengths:**

The benchmark is built upon a dataset of real user interactions (WildChat), ensuring the queries reflect genuine spoken language use cases and user behavior.

The benchmark systematically incorporates paralinguistic phenomena and acoustic challenges that are largely ignored by text-centric benchmarks, providing a more comprehensive test of model capabilities.

**Weaknesses:**

1. The benchmark is relatively small (1100 queries, with only 100 for paralinguistic features) and relies heavily on synthesized speech from only two base speakers.

2. The benchmark focuses exclusively on single-turn interactions, omitting the critical aspect of multi-turn conversations, which is a fundamental element of conversational AI and a key use case for voice assistants.


3. Both the dataset curation and the core evaluation protocol depend heavily on another LLM (GPT-4o-mini). This replaces rigorous scientific validation with automated prompts, raising serious concerns about the reliability, accuracy, and circularity of the benchmark's judgments.

**Questions:**

Overall, the existing evaluation data is too limited and has many sub-categories, thus slightly data leakage could make the benchmark invalid. Is there a efficient way to generate more data to make the benchmark more solid?

---

> ### Author Response · Authors · 2025-11-21
> **Author Response to Reviewer DAeh (Part 1/2)**
>
> We thank Reviewer DAeh for the thoughtful feedback and rating. We would like to address the key concerns regarding dataset scale, TTS reliance, single-turn focus, and evaluation circularity.
>
> ## On Weakness 1 (Scale/TTS) & Question 1 (Data Generation/Leakage)
> We will address the reviewer's points regarding scale, data generation, and TTS reliance together, as they are related.
>   1. **On Data Generation (Q1) & Practicality of Scale (W1):** The reviewer asks about efficiently generating more data. The primary bottleneck is **not** data generation, but the **extreme cost and slowness of S2S evaluation**. S2S evaluation faces unique challenges distinct from text-based evaluation, making it exceptionally costly:
> 	  - **Slow S2S Inference:** Models are complex (e.g., GLM-4-Voice's interleaved strategy, Qwen-2.5-omni's 'talker' module).
> 	  - **Slow Vocoder Step:** A separate, computationally intensive vocoder is required to convert inferred tokens into a final waveform.
> 	  - **Slow Evaluation Pipeline:** The resulting audio must then undergo a *two-pass evaluation*: (a) a full **ASR transcription pass** for semantic scoring , plus (b) a separate **acoustic quality assessment** (e.g., UTMOS).
>
>       This entire pipeline makes a massive benchmark **unusable** for most researchers. This directly answers Q1: a more "efficient" automated generation is not practical because the *evaluation* cost is the bottleneck. Therefore, we **"prioritized diagnostic depth over sheer volume"** .
>
>    Besides, the reviewer correctly notes that with a small set, we must prove the results are robust. To address this, we **conducted paired t-tests** on our results. This analysis confirms that our 1.1k-sample set, while focused, is **statistically sufficient** to distinguish model performance, and our findings are not an artifact of data leakage or random noise.
> 	*(P-value for H_a: Row Model Score > Column Model Score, based on Overall Avg. from Table 2)*
>
> | | Naive Pipeline (5.24) | Kimi-Audio (4.54) | GLM-4-Voice (5.03) | MiniCPM (5.08) | Qwen-2.5-omni (6.01) | GPT-4o-Audio (6.29) |
> | :--- | :---: | :---: | :---: | :---: | :---: | :---: |
> | **Naive Pipeline (5.24)** | - | **<0.01*** | **<0.01*** | **<0.01*** | p>0.99 | p>0.99 |
> | **Kimi-Audio (4.54)** | p>0.99 | - | p>0.99 | p>0.99 | p>0.99 | p>0.99 |
> | **GLM-4-Voice (5.03)** | p>0.99 | **<0.01*** | - | p=0.83 | p>0.99 | p>0.99 |
> | **MiniCPM (5.08)** | p>0.99 | **<0.01*** | p=0.17 | - | p>0.99 | p>0.99 |
> | **Qwen-2.5-omni (6.01)**| **<0.01*** | **<0.01*** | **<0.01*** | **<0.01*** | - | p>0.99 |
> | **GPT-4o-Audio (6.29)** | **<0.01*** | **<0.01*** | **<0.01*** | **<0.01*** | **<0.01*** | - |
>   ##  On TTS Reliance & "Two Base Speakers" (Weakness1）
>   We respectfully clarify **a key misunderstanding** regarding our speaker diversity. Contrary to the premise that we relied on "only two base speakers," our benchmark employs **a robust, hybrid strategy** designed to maximize both scale and realism:
> - **Expanded Speaker Diversity:** As detailed in Section 2.1.2, we curated **16 distinct voice profiles**. We achieved this by cloning unique timbres across **four distinct age groups** and **two genders**, ensuring the models are tested against a wide variety of vocal characteristics rather than a single pair of base voices.
> - **Strategic Human Recordings:** We fully agree that TTS can be insufficient for nuanced speech. Therefore, for the 100-sample Paralinguistic-Featured (PF) set—where prosody and emotion are paramount—we prioritized authenticity by using 100% direct human recordings .
> - **External Validation:** The validity of using TTS for the broader set of general queries is supported by concurrent research. VoiceBench [1]  explicitly studied this variable, finding that model rankings remain **"consistent"** and show a **"strong correlation"** when evaluated on real vs. synthetic speech.
> - **Bridging the Realism Gap:** Finally, we addressed the primary difference between TTS and real-world audio—cleanliness—by systematically augmenting *all* samples with realistic **"Human Noise"** and **"Natural Noise"** . This ensures our synthetic data presents a genuine acoustic challenge.
>
> **Due to character limitations, our response continues in the following comment.**

---

> ### Author Response · Authors · 2025-11-21
> **Author Response to Reviewer DAeh (Part 2/2)**
>
> Continuing from the previous comment, here we provide our responses to the rest questions.
>
> ## **On Weakness 2 (Single-Turn Interactions)**
> This was an **explicit design choice**, not an omission. We note that **current mainstream speech benchmarks** (e.g., VoiceBench, AIR-Bench) also predominantly focus on single-turn interactions, as this remains a highly complex and unsolved area.
>
> Our goal with WildSpeech-Bench is not to solve **every** open problem, but to address the **most urgent and previously neglected ones**: establishing a solid baseline for single-turn robustness to noise, speaker diversity, and paralinguistics. As detailed in **Appendix J.1**, meaningful multi-turn evaluation **cannot** be built upon flawed single-turn foundations. Our work establishes this necessary solid foundation.
> ## **On Weakness 3 (Evaluation Circularity)**
> We respectfully clarify that our method does not "replace" rigorous validation; rather, it **standardizes** it through a verifiable, reproducible, and human-anchored protocol. We address the specific concerns of **circularity**, **accuracy**, and **reliability** with four layers of defense:
> 1. **Defense Against Circularity (New Experiment):** The reviewer correctly points out the risk of "self-preference bias" (OpenAI judge favoring OpenAI models). To empirically test this, we **conducted a new experiment** by re-scoring all models using a judge from a completely different model family: **Qwen2.5-7B**.
> 	- **Hypothesis:** If the benchmark results were driven by circularity/bias, the non-OpenAI judge should produce a significantly different ranking.
> 	- **Result:** As shown in the table below, while the absolute score distribution shifts (Qwen is generally more generous), the **relative ranking of the models remains identical**. This empirically proves that the performance gap is intrinsic to the models' capabilities, not an artifact of the judge's origin.
> 	**Judge Robustness Analysis: GPT-4o-mini vs. Qwen-Judge**
> 	| Model | Score (GPT-4o-mini Judge) | Score (Qwen-Judge) | Rank (GPT-4o-mini) | Rank (Qwen-Judge) |
> 	| :--- | :---: | :---: | :---: | :---: |
> 	| GPT-4o-Audio | 6.29 | 6.82 | **1** | **1** |
> 	| Qwen-2.5-omni | 6.01 | 6.71 | **2** | **2** |
> 	| Naive Pipeline | 5.24 | 6.21 | **3** | **3** |
> 	| MiniCPM | 5.08 | 5.79 | **4** | **4** |
> 	| GLM-4-Voice | 5.03 | 5.58 | **5** | **5** |
> 	| Kimi-Audio | 4.54 | 5.12 | **6** | **6** |
> 2. **Defense of Accuracy (Human Anchoring):** The ultimate measure of scientific validity is alignment with ground truth. Our automated scores achieve a **high Pearson correlation (r=0.86)** with **human expert evaluators** . Since our human experts are unbiased, this strong alignment confirms that our automated pipeline serves as a highly accurate proxy for human perception.
> 3. **Defense of Reliability (The Checklist Mechanism):** We must emphasize that we are not using generic, subjective prompts (e.g., "Rate this response"). Our **Query-Aware Checklist** (Appendix H) transforms the evaluation from a **"subjective critique"** task into an **"objective information extraction"** task. By forcing the judge to verify the presence of specific facts and logical steps defined in the checklist, we structurally minimize the hallucination and subjectivity often associated with LLM evaluators.
> 4. **External Validation:** Finally, our findings align with broader research on LLM-as-a-judge [2,3] , which consistently demonstrates that strong LLMs, when constrained by clear rubrics (as in our checklists), provide objective assessments that align with human preferences.
>
>
> We stand ready to answer any follow-up questions immediately. We hope that these clarifications effectively address your concerns and **would appreciate your reconsideration of the paper's contribution.**
>
>
> ## reference
> 1. Chen, Yiming, et al. "Voicebench: Benchmarking llm-based voice assistants."
> 2. Qingquan, Li et,al. "Evaluating Scoring Bias in LLM-as-a-Judge"
> 3. Yusuke, Yamauchi et, al. "An Empirical Study of LLM-as-a-Judge:How Design Choices Impact Evaluation Reliability"

---

### Official Review · Reviewer_sSJC · 2025-11-01

**Soundness:** 2
**Presentation:** 3
**Contribution:** 2
**Rating:** 4
**Confidence:** 4

**Summary:**

The paper introduces WildSpeech-Bench, a benchmark tailored for evaluating end-to-end SpeechLLMs (speech-to-speech models). Unlike prior S2T-oriented benchmarks (e.g., VoiceBench, OpenAudio, SD-Eval), this work constructs a dataset of 1,100 curated speech queries spanning five categories (Information Inquiry, Solution Request, Opinion Exchange, Text Creation, and Paralinguistic-Featured). It employs both synthetic and human-recorded audio, includes multi-tiered noise augmentation, and proposes a query-aware evaluation framework that improves automatic alignment with human judgments. Experimental results compare major proprietary and open-source models (GPT-4o-Audio, Qwen2.5-Omni, GLM-4-Voice, etc.), showing that current systems degrade sharply under noise and paralinguistic variation.

**Strengths:**

1. The paper propose the evaluation of end-to-end SpeechLLMs, which are rapidly emerging but lack standardized benchmarks.
2. The benchmark construction is well-motivated and systematic, combining real user queries from WildChat, voice cloning, paralinguistic diversity, and noise realism.
3. The use of customized, query-specific rubrics improves correlation with human judgments (Pearson r = 0.86), a clear methodological improvement over prior automatic evaluations.
4. The comparison of multiple leading models and ablation studies (multi-round ASR, noise effects, etc.) are thorough and informative.
The reproducibility and LLM-usage statements are commendable. Figures and tables are well organized.

**Weaknesses:**

1. The dataset (1.1 k samples) is small relative to modern benchmarks. While quality is prioritized, the paper lacks analysis of statistical coverage or how well this scale generalizes across domains or accents. Expansion beyond English is critical.
2. Despite inclusion of 100 human recordings, most data come from TTS (CosyVoice), which constrains prosodic and emotional realism. Results may overestimate model robustness to human variability.
3. Although the authors mitigate transcription bias via multi-round ASR and query-aware scoring, evaluation still depends on text-based scoring. A truly end-to-end audio–audio metric (e.g., perceptual similarity or content alignment) is absent.
4. Potential circularity in LLM-based evaluation. The benchmark uses GPT-4o-mini as the judge, while GPT-4o-Audio is also a tested model—raising potential bias in favor of OpenAI systems. The paper should quantify or control for evaluator–model overlap.
5. Limited discussion of error analysis and ethical aspects. While a few case studies are shown, the error taxonomy is shallow. No discussion of dataset bias (e.g., cultural, gender, or accent representation) or privacy in using WildChat data.
6. While human scoring is mentioned, the setup (e.g., annotator consistency, inter-rater reliability, audio listening conditions) is relegated to Appendix B and not summarized quantitatively.

**Questions:**

1. Were the differences in Table 2 tested for significance (e.g., paired t-tests or bootstrap confidence intervals)?
2. Without statistical testing, how confident can we be that GPT-4o-Audio’s reported advantage is not due to noise in small-sample evaluation?
3. Did you analyze whether performance degradation under noise correlates with specific frequency bands or noise types (e.g., human vs. ambient)?
4. Would fine-grained error analysis help isolate which components of end-to-end models (encoder vs. decoder) are most vulnerable?

---

> ### Author Response · Authors · 2025-11-21
> **Author Response to Reviewer sSJC (Part 1/3)**
>
> We thank Reviewer sSJC for the thorough review and constructive feedback. We address all weaknesses and questions below.
> ## On Weakness 1  and Questions 1 & 2 (Scale & Statistical Significance)
> We address these related points together, as they are central to our paper's validation.
>   1. **On Scale:** We must clarify that the 1.1k scale is a **necessary and deliberate choice**. S2S evaluation faces **unique and severe practical bottlenecks** distinct from text-based evaluation, making it exceptionally costly to run:
>
>      - **Slow S2S Inference:** S2S models are complex (e.g., GLM-4-Voice's interleaved strategy [1], Qwen-3-omni's 'talker' module  [2]).
>      -  **Slow Vocoder Step:** A separate, computationally intensive vocoder (e.g., flow matching + HiFi-GAN) is required to render the waveform.
>      - **Slow Evaluation Pipeline:** The resulting audio must then undergo a full ASR transcription pass (e.g., Whisper-large-v3, as described in Section 2.2 ) *plus* a separate acoustic quality assessment (e.g., UTMOS, Section 2.2) .
>
>      This entire pipeline makes a massive benchmark **unusable** for most researchers. We therefore "prioritized diagnostic depth over sheer volume".
>
> 2. **On Statistical Significance:** **Following your constructive suggestion** to verify that our results are not due to small-sample noise, we have conducted **paired t-tests** on our 1,100 evaluation samples.
> *(P-value for H_a: Row Model Score > Column Model Score, based on Overall Avg. from Table 2)*
> | | Naive Pipeline | Kimi-Audio | GLM-4-Voice | MiniCPM | Qwen-2.5-omni | GPT-4o-Audio |
> | :--- | :---: | :---: | :---: | :---: | :---: | :---: |
> | **Naive Pipeline** | - | **<0.01*** | **<0.01*** | **<0.01*** | p>0.99 | p>0.99 |
> | **Kimi-Audio** | p>0.99 | - | p>0.99 | p>0.99 | p>0.99 | p>0.99 |
> | **GLM-4-Voice** | p>0.99 | **<0.01*** | - | p=0.83 | p>0.99 | p>0.99 |
> | **MiniCPM** | p>0.99 | **<0.01*** | p=0.17 | - | p>0.99 | p>0.99 |
> | **Qwen-2.5-omni** | **<0.01*** | **<0.01*** | **<0.01*** | **<0.01*** | - | p>0.99 |
> | **GPT-4o-Audio** | **<0.01*** | **<0.01*** | **<0.01*** | **<0.01*** | **<0.01*** | - |
>
> Our analysis confirms that our key findings are statistically robust. For example, the advantage of GPT-4o-Audio over Qwen-2.5-omni is **statistically significant** (p < 0.01). The analysis also demonstrates the fidelity of our benchmark by correctly identifying which models perform statistically similarly (e.g., the minor 0.05-point difference between GLM-4-Voice and MiniCPM is not statistically significant, p=0.17), confirming they belong to the same performance tier.
>
> ##   On Weakness 2 (TTS Realism)
> The core question raised is whether the use of synthesized speech (TTS) introduces a bias that distorts the evaluation outcome compared to real human speech. Evidence from concurrent research and our specific design confirms that our methodology is valid and reliable.
> 1. **External Validation:** The VoiceBench [3] paper rigorously studied this exact issue by comparing model performance on real versus synthetic versions of the same dataset. Their findings confirm that model rankings remain **"consistent"** and show a **"strong correlation"** regardless of the audio source. This strongly indicates that using high-quality TTS is a scientifically valid proxy for benchmarking general capabilities.
> 2. **Methodological Rigor:** We did not rely on a single generic voice. We systematically addressed the limitations of TTS through a three-pronged approach:
> 	- **Diversity:** We utilized 16 distinct voice profiles (cloning diverse age and gender groups) to ensure models are tested against varied timbres.
> 	- **Acoustic Realism:** We augmen *all* samples with realistic "Human Noise"* and "Natural Noise".
> 	- **Paralinguistic Nuance:** For the specific subset where TTS fails to capture essential prosody (the Paralinguistic-Featured set), we used 100% direct human recordings.
>
> ##  Reference
> 1. Zeng, Aohan, et al. "Glm-4-voice: Towards intelligent and human-like end-to-end spoken chatbot."
> 2. Jin Xu, et, al. Qwen3-Omni Technical Report
> 3. Chen, Yiming, et al. "Voicebench: Benchmarking llm-based voice assistants."
> 4. Qingquan, Li et,al. "Evaluating Scoring Bias in LLM-as-a-Judge"
> 5.  Yusuke, Yamauchi et, al. "An Empirical Study of LLM-as-a-Judge:How Design Choices Impact Evaluation Reliability"
>
>
>
> **Due to character limitations, our response continues in the following comment.**

---

> ### Author Response · Authors · 2025-11-21
> **Author Response to Reviewer sSJC (Part 2/3)**
>
> Following up on our discussion in Part 1, we address Weakness 3 and Weakness 4 below.
>
> ##  On Weakness 3 (Evaluation depends on text-based scoring)
>   We address the reviewer's suggestion for a "truly end-to-end audio-audio metric" (e.g., perceptual similarity). We respectfully argue that such metrics are **methodologically unsuitable** for open-ended SpeechLLM benchmarks, which necessitated our specific design choice.
> 1. **Impracticality of Audio-Audio Metrics:** Metrics like perceptual similarity or waveform alignment require a **"golden reference" audio** to compare against. In a generative dialogue setting, defining a "golden truth" for timbre or prosody is impossible. Different models have legitimate, distinct speaker identities (e.g., GLM-4-Voice has a **fixed, fine-tuned voice** ). Using an audio-audio metric would unfairly penalize a model simply for having a different voice or prosodic style than the reference, even if its output is perfect.
> 2. **Our Solution (Decoupled Evaluation):** Because a single end-to-end metric cannot fairly capture both content and diverse speaker qualities, we adopted a decoupled, dual-track design principles:
>     -  **Semantic Evaluation (Track 1):** We use ASR as a *tool* to extract content for semantic scoring. To ensure robustness against the transcription errors noted in Appendix E, we deliberately avoided brittle formats like MCQs and used open-ended queries. Critically, we empirically validated this approach against **human experts who evaluated the raw audio directly**. The high correlation (Pearson r=0.86) between our ASR-based scores and human scores confirms that our pipeline effectively mitigates transcription errors and aligns with human perception .
>     -   **Audio Quality Evaluation (Track 2):** We assess acoustic and paralinguistic quality directly on the generated audio, completely bypassing ASR. This is achieved through UTMOS scores, ensuring that speech-centric qualities are evaluated in their native modality.
> ##  On Weakness 4 (Evaluation Bias)
> We thank the reviewer for raising this critical point regarding potential evaluator circularity. We agree that quantifying and controlling for this overlap is essential. We addressed this through a new experiment and multi-layered validation:
> 1. **Control for Evaluator Overlap (New Experiment):** To strictly control for potential bias in favor of OpenAI systems, we **conducted a new experiment** as requested. We re-scored all models using a powerful, non-OpenAI judge (**Qwen2.5-7B**). As shown in the table below, while the absolute scoring distribution differs, the **relative ranking of the models remains identical**. This empirically proves that our results are robust and reflect true model capabilities rather than a specific judge's preference.
> 	**Judge Robustness Analysis: GPT-4o-mini vs. Qwen-Judge**
> 	| Model | Score (GPT-4o-mini Judge) | Score (Qwen-Judge) | Rank (GPT-4o-mini) | Rank (Qwen-Judge) |
> 	| :--- | :---: | :---: | :---: | :---: |
> 	| GPT-4o-Audio | 6.29 | 6.82 | **1** | **1** |
> 	| Qwen-2.5-omni | 6.01 | 6.71 | **2** | **2** |
> 	| Naive Pipeline | 5.24 | 6.21 | **3** | **3** |
> 	| MiniCPM | 5.08 | 5.79 | **4** | **4** |
> 	| GLM-4-Voice | 5.03 | 5.58 | **5** | **5** |
> 	| Kimi-Audio | 4.54 | 5.12 | **6** | **6** |
> 2. **External Evidence:** This result aligns with established research on LLM-as-a-judge [3,4], which shows that strong LLM judges, when well-prompted with clear rubrics, are generally objective and align well with human preferences regardless of their origin.
> 3. **Defense by Mechanism (Checklists):** Our framework is explicitly designed to minimize subjectivity. Our **Query-Aware Checklist** (Appendix H, Table 7) provides fact-based rubrics rather than subjective prompts. This forces the judge to evaluate *objective correctness* (facts, steps) over *style*, neutralizing "style bias."
> 4. **Validation against Humans:** Finally, the ultimate validation is our **high (0.86) Pearson correlation** with **human expert evaluators**. Since our human experts are unbiased, the strong alignment between them and our automated judge confirms the evaluation is reliable.
>
>
> Due to character limitations, our response continues in the following comment.

---

> ### Author Response · Authors · 2025-11-21
> **Author Response to Reviewer sSJC (Part 3/3)**
>
> Continuing from the previous comment, here we provide our responses to the rest questions.
>
> ##  On Weakness 5 (Error Analysis & Ethics)
>   - **On Error Analysis (W5):** You are correct that our initial case studies in **Section 3.6**  were illustrative rather than comprehensive. To address the "shallow taxonomy" and provide a deeper analysis, we have **conducted a new quantitative error analysis**. We manually categorized a sample of "bad cases" (defined as responses scoring 1-2 points) to identify the most common failure modes for SOTA end-to-end models on our benchmark.
> This new analysis reveals that errors in the speech modality are diverse. We found that **Proper Noun Misunderstanding** (including long-tail names, abbreviations, and homophones) is a major challenge specific to speech models, accounting for over a third of significant errors.
> Appendix Table Z: Quantitative Taxonomy of "Bad Case" Errors (Scores 1-2)
> | Failure Mode | Distribution |
> | :--- | :---: |
> | **Proper Noun Errors (Total)** | **33.9%** |
> |    ↳ *General Proper Noun* | (19.4%) |
> |   ↳ *Abbreviation* | (9.7%) |
> |    ↳ *Near-Homophone* | (4.8%) |
> | **Irrelevant Response** | 30.6% |
> | **Language Confusion** | 14.5% |
> | **Factual Error** | 9.7% |
> | **Unwarranted Refusal** | 9.7% |
> | **Hallucination / Nonsense** | 1.6% |
> | **Total** | **100.0%** |
>   - **Gender/Age:** We will explicitly discuss the demographics of our speakers. Our 1,000 general queries were synthesized using **16 distinct voice profiles** (cloned from male/female speakers across four age groups: child, adolescent, adult, senior) as detailed in **Section 2.1.2** . Our 100 PF queries were recorded by one male and one female native speaker.
>   - **Privacy (WildChat):** We will clarify our data handling protocol. Our work is based on the public, anonymized **WildChat** dataset. We strictly adhered to its terms of use, and our own multi-stage manual curation and verification process served as an additional safeguard to review content and ensure no PII was present.
> ##  On Weakness 6 (Human Scoring Details)
> We thank the reviewer for this fair point regarding rigor and presentation. Our human evaluation serves as the gold standard for our benchmark, and we agree that its setup and reliability should be summarized in the main paper. This information was relegated to **Appendix B**  in our initial submission **due to the strict page limit**.
>   1. **Audio Listening Conditions:** We will clarify the missing setup details. All annotations were performed in a **quiet, controlled office environment**. Annotators were required to use **high-fidelity, closed-back headphones** to ensure that subtle acoustic cues (especially from our noise-augmented data and the Paralinguistic-Featured set) were clearly perceived without interference.
>   2. **Rigorous Consensus Protocol:** Our reliability is ensured not by averaging independent scores, but by a multi-stage protocol designed to achieve expert consensus, as detailed in **Appendix B** . This protocol included:
> 	  - **Expert Annotators:** A team of six university graduates: four annotators with over one year of experience in LLM evaluation, and two senior Quality Control (QC) specialists with over two years of specific training .
> 	  - **Two-Stage 100% Review:** A strict protocol where (1) annotators performed initial scoring, and (2) our QC specialists **reviewed 100% of all annotations** for consistency and accuracy .
> 	  - **Calibration & Discussion:** A mandatory discussion-based arbitration for ambiguous or complex cases. The final score was *only* determined after this joint discussion between the annotators and the QC group to ensure a fair and calibrated standard .
> 	  - **Identical Guidelines:** Annotators and QC specialists used the *exact same* detailed guidelines and query-aware checklists that were provided to our automatic evaluation system .
> We will **move this complete summary** of the setup (including the listening conditions and the two-stage consensus protocol) from Appendix B into **Section 3.3** to improve transparency and fully validate the robustness of our human baseline.
> ## On Question 3 (Noise Type Correlation)
> This is an excellent question. A preliminary analysis of our **Figure 5** data shows that **"Human Noise"** causes a more significant performance degradation across all models than "Natural Noise" . This suggests models struggle more with the "cocktail party problem" (isolating speech-from-speech) than with general robustness to ambient sound.
> ## On Question 4 (Encoder vs. Decoder Failure)
> This is a valuable research question. Our benchmark is designed to evaluate the *end-to-end* system's performance. Isolating which specific component (encoder vs. decoder) is the primary point of failure would require specialized, model-specific ablations that are beyond the scope of a general benchmark. We agree this is a critical next step for *model developers* using our benchmark and will add this to our Future Work.

---

### Official Review · Reviewer_akXR · 2025-11-01

**Soundness:** 3
**Presentation:** 3
**Contribution:** 3
**Rating:** 6
**Confidence:** 4

**Summary:**

The authors introduce WildSpeechBench, a benchmark to evaluate spoken language models (SLMs) on realistic audio. The benchmark is developed by extensively filtering human-AI conversations and synthesizing them into audio with a TTS system. To better reflect real-world audio, acoustic noise is added to each synthetic speech sample. The authors complement the synthetic speech by recording real human voice actors on a subset of the data. Finally, the authors benchmark a variety of SLMs on the data, showing that leading open-source models significantly lag behind proprietary ones.

**Strengths:**

- The authors develop a pipeline to create test samples with life-like synthetic speech by combing TTS and noise
- The authors complement the synthetic samples with real recorded speech
- The authors show that SOTA SLMs exhibit significantly degraded performance when processing samples with noise, showing their fragility despite strong reported results

**Weaknesses:**

- The set of benchmarked systems is relatively small, missing notable models like Moshi
- My main concern is the lack of evaluation on the realism of the synthetic noisy audio. All of the results of this paper are dependent on this factor. Since real speech is already used for a subset, why not synthesize another small subset with the real-speech as the speaker prompt? Then you can have a direct comparison of the impact of real speech vs synthetic noisy samples.

**Questions:**

>  revealing that strong performance on S2T dialogue benchmarks does not
translate to S2S settings

I believe the benchmarked models all support text output. In that case, obtaining S2T versions of the model outputs should be feasible. I think it would be beneficial to have a comparison between the S2T results and S2S results to rigorously validate this claim

---

> ### Author Response · Authors · 2025-11-21
> **Author Response to Reviewer akXR**
>
> We thank Reviewer akXR for the positive assessment of our work's contributions and for the insightful, constructive questions.
> ## Regarding Weakness 1
> To address this weakness and ensure our benchmark captures the latest developments, we have significantly **expanded our evaluation**. Per your suggestion, we added **Moshi**. Furthermore, to provide a comprehensive view of the open-source landscape, we also evaluated **Mini-Omni [1]**, **Mini-Omni-2 [2]**, and the newly released **MiMo-Audio [3]**.
> | Model | TC | II | SR | OE | PF | Overall Avg. |
> | :--- | :---: | :---: | :---: | :---: | :---: | :---: |
> | **Moshi (new)** | 1.11 | 1.15 | 1.28 | 1.36 | 1.41 | 1.24 |
> | **Mini-Omni (new)** | 1.91 | 2.22 | 2.65 | 2.39 | 1.67 | 2.26 |
> | **Mini-Omni-2 (new)** | 2.47 | 2.51 | 3.17 | 3.31 | 2.41 | 2.75 |
> | Naive Pipeline | 5.55 | 4.98 | 5.51 | 5.18 | 4.84 | 5.24 |
> | Kimi-Audio | 4.45 | 4.33 | 4.79 | 4.70 | 4.92 | 4.54 |
> | GLM-4-Voice | 5.16 | 4.77 | 5.41 | 5.04 | 4.51 | 5.03 |
> | MiniCPM | 5.17 | 4.89 | 5.28 | 5.31 | 4.78 | 5.08 |
> | **MiMo-Audio (new)** | 6.46 | 5.44 | 6.07 | 5.81 | 5.43 | 5.77 |
> | Qwen-2.5-omni | 5.98 | 5.84 | 6.66 | 6.16 | 4.46 | 6.01 |
> | GPT-4o-Audio | 6.74 | 6.06 | 6.39 | 6.32 | 6.01 | 6.29 |
>
> As requested, we integrated Moshi into our pipeline. However, as shown in the table, its performance on our generative instruction-following tasks is limited (Overall Avg: 1.24). We attribute this to Moshi's architectural focus on low-latency, full-duplex streaming rather than complex reasoning or knowledge retrieval. This finding is **consistent with other public benchmarks**, such as the VoiceBench leaderboard [4], where Moshi similarly trails behind other models. Besides, the newly added MiMo-Audio demonstrates strong performance, further validating the benchmark's ability to distinguish between model capabilities.
> ## Regarding Weakness 2
> We thank the reviewer for this insightful and constructive suggestion. The proposed ablation study—directly comparing model performance on real speech versus synthetic speech—is indeed the definitive way to verify that benchmark results are not artifacts of the synthesis process.
> We address this fundamental validity question by drawing on robust evidence from concurrent research and our own specific design:
> 1. **Evidence from Concurrent Research (VoiceBench):** This exact comparison **has been rigorously analyzed** in the recent VoiceBench [4] study. They performed the specific ablation suggested by the reviewer (comparing real vs. synthetic versions of the same dataset) and their findings provide a clear answer: model rankings remain **"consistent"** and show a **"strong correlation"** regardless of whether the audio is real or synthesized . These findings **strongly support the validity** of using high-quality, diverse TTS (our benchmark uses 16 distinct profiles) as a reliable proxy for evaluating general model capabilities.
> 2. **Strategic Human Recordings:** To further ensure authenticity where it matters most, we went a step beyond general TTS. For the **Paralinguistic-Featured (PF)** set—where prosodic nuance is critical and TTS is most likely to fail—we used **100% direct human recordings**.
>
> By combining the statistically validated reliability of TTS for general content with genuine human recordings for paralinguistic nuances, our hybrid approach ensures both scalability and "in-the-wild" fidelity.
> ## Regarding Questions 1
> We appreciate the reviewer's insightful suggestion to directly compare S2T and S2S outputs. Following your recommendation, **we conducted this specific comparison** to rigorously validate the claim that S2T performance does not guarantee S2S success.
> We selected Kimi-Audio for this new experiment because it exhibits exceptional performance on established S2T benchmarks (such as VoiceBench), making it the ideal candidate to test whether that strength transfers to speech generation.
> | Metric | Overall Average Score | TC (Text Creation) | II (Information Inquiry) | SR (Solution Request) | OE (Opinion Exchange) | PF (Paralinguistic) Avg. |
> | :--- | :---: | :---: | :---: | :---: | :---: | :---: |
> | S2T  | 6.64 | 6.80 | 6.49 | 7.09 | 6.47 | 5.41 |
> | S2S  | 4.54 | 4.45 | 4.33 | 4.79 | 4.70 | 4.92 |
> | Difference | -2.10 | -2.35 | -2.16 | -2.30 | -1.77 | -0.49 ||
>
> As the table clearly illustrates, Kimi-Audio's overall performance drops by **-2.10 points** when utilizing its S2S pipeline, with the most severe degradation occurring in factual generation tasks (TC and SR).
>
> ## Reference
>   1. Xie, Zhifei, et al. "Mini-omni: Language models can hear, talk while thinking in streaming."
>   2. Xie, Zhifei, and Changqiao Wu. "Mini-omni2: Towards open-source gpt-4o with vision, speech and duplex capabilities."
>   3. Cici, et al. MiMo-Audio: Audio Language Models are Few-Shot Learners
>   4. Chen, Yiming, et al. "Voicebench: Benchmarking llm-based voice assistants."

---

### Meta-Review · Area_Chair_ujvc · 2026-01-02

**Summary:**

Despite the authors claiming that three reviewers ''(akXR, DAeh, rPtm) consistently acknowledged the paper’s contribution in filling the S2S evaluation gap and the novelty of the noise-augmented pipeline,''  the reviewers appeared to firmly agree on one point: the proposed benchmark is limited in many ways. For example, akXR states that “the set of benchmarked systems is relatively small”; sSJC commented that “the dataset (1.1k samples) is small relative to modern benchmarks”; DAeh noted that “the benchmark is relatively small (1,100 queries, with only 100 for paralinguistic features) and relies heavily on synthesized speech from only two base speakers”; and PXjr observed that “the dataset contains only 1,100 queries in total. This scale is insufficient for a robust and generalizable benchmark.”

**Reviewer Concerns:**

The authors have addressed some of the reviewers’ concerns, such as (i) the lack of statistical significance tests, (ii) evaluation dependency on text-based scoring, and (iii) potential circularity in LLM-based evaluation. Yet, it appears that key concerns that limit the superiority of this benchmark over previously existing ones are still outstanding: (i) the small size of the benchmark, (ii) its limitation to English, and (iii) the benchmark being 100% focused on single-turn evaluation, similar to previously existing benchmarks.

**Reviewer Scores:**

The reviewers might have left the score unchanged, or maybe one of them (the only reviewer, namely rPtm, most positive about the proposed work) might have increased their score.

---

### Decision · Program_Chairs · 2026-01-26

Reject